

# Life cycle of a flower cloud system during the EUREC⁴A field campaign

Zhiqiang Cui[1,2], Alan Blyth[1,3], Ralph Burton[3], Sandrine Bony[4], Steven Böing[1], Alan Gadian[1], and Leif Denby[1]

[1]Institute for Climate and Atmospheric Science,  School of Earth and Environment, University of Leeds, Leeds, LS2 9JT, UK
[2]Centre for Environmental Modelling and Computation (CEMAC), University of Leeds, Leeds, LS2 9JT, UK
[3]National Centre for Atmospheric Science, Leeds, LS2 9PH, UK
[4]LMD/IPSL, Sorbonne Université, CNRS, Paris, France

*Correspondence to*: Zhiqiang Cui (z.cui@leeds.ac.uk)

**Abstract.** The organisation of trade-wind cumulus clouds in the vicinity of Barbados can affect the amount and lifetime of the clouds thereby potentially playing an important role in the top of the atmosphere radiation budget. This paper presents a case study of a flower cloud system that occurred on 2 February 2020 near Barbados during the EUREC⁴A field campaign. The evolution of a cluster of clouds that developed into a flower system is investigated from 0000 UTC to 2300 on 2 February using GOES-16 satellite IR and visible images, dropsonde data, and ERA5 reanalysis data. The cloud system began as small clouds ~ 400 km to the north-east of Barbados. Aggregation of clouds continued to occur as the system moved towards Barbados. A striking feature was the development of a large area of rain in the central region of the system in the later stages of development when the area plateaued. Several cloud arcs associated with cold pools became visible as they emerged from the cloud shield. The flower system began to decay about 7 hours after the maximum values of derived effective radius. The environmental conditions determined from ERA5 reanalysis along the trajectory of the flower were examined. The sea surface temperatures were 26.9 ± 0.3°C along the trajectory. The surface wind speed in the proximity of the flower increased during the first six hours in the early development stage from 7 m s$^{-1}$ to 9 m s$^{-1}$ and then decreased gradually to about 5 m s$^{-1}$ during the next 12 hours. The inversion strength measured with dropsonde was greater in the flower than in its surrounding locations due to the mesoscale variation in temperature field and, probably, the interactions between the flower and the atmosphere. The cloud system was surrounded by high total precipitable water in the early stages and along the trajectory as it developed into the flower. The Barbados region was marked by high aerosol optical depth on the day. The life cycle of the flower cloud system characterized in this paper will help evaluate numerical simulations of shallow convective organisation.



## 1 Introduction

Elucidating the role of cloud-circulation coupling in climate (EUREC$^4$A) is an international project that included a field study of trade-wind cumulus in the Barbados region (Bony et al., 2017; Stevens et al., 2021). Prior to the field campaign, Stevens et al. (2019) revealed that four patterns of mesoscale organization of cumulus exist in the area east of Barbados from

48°W to 58°W, 10°N to 20°N. Those four organizations are labelled as sugar, gravel, fish, and flower. Sugar is composed of scattered, low clouds; gravel clouds are accompanied by evident gust fronts; fish are elongated cloud system with a fishbone-like structure; and flowers are round flower-like features. Bony et al. (2020) further investigated the dependence of the four patterns on environmental conditions by analysing 19 years of satellite data. They found two dominant factors: the near-surface wind speed and the strength of the lower-tropospheric stability. Sugar and gravel tend to occur in unstable

environments. Gravel occurs in windier conditions ($v_s \sim 9$ m s$^{-1}$), while sugar in calmer conditions ($v_s \sim 6 - 7$ m s$^{-1}$). Although fish and flowers, unlike sugar or gravel, tend to occur in stable environments, fish likely occurs in weaker winds ($v_s \sim 6 - 7$ m s$^{-1}$), and flowers in stronger winds ($v_s \sim 9$ m s$^{-1}$).

Previous studies have shown the factors affecting cloud organisation. Hill (1974) investigated the size and spacing of

cumulus clouds using a numerical model and found five controlling factors: the suppressing effect of larger neighboring cloud, environmental warming, eddy mixing, the presence of an inversion above, and downdraughts associated with precipitation. Further, it was also noticed that the key factor ruling the development of large multi-cell clouds was the inclusion of surrounding moisture anomalies. Lock (2009) numerically studied the processes affecting the cumulus cloud area and identified additional factors that are important for the development of the clouds, such as the strength of the surface

fluxes, which also affects the strength of the cumulus mass transport, the moisture content of the free atmosphere and the strength of cloud-top radiative cooling.

Klein et al. (2017) made a thorough review of low-cloud controlling factors and listed seven controlling factors. The first is strengthened inversion stability, which reduces mixing across inversion keeps boundary layer shallower, more humid and

cloudier. The second factor is reduced subsidence, which increases boundary layer depth, hence, cloud. The third is related to increased horizontal cold advection, which destabilizes the surface–atmosphere interface and increases upward buoyancy flux, and promotes more clouds. The fourth factor is increased free-tropospheric humidity. Therefore, a more humid troposphere reduces entrainment drying, thus, increases cloud. The fifth one is decreased downward longwave radiation. The reduced downward longwave radiation increases cloud-top radiative cooling, drives more turbulence and supports cloud. The

sixth factor is colder sea-surface temperature (SST), which reduces the efficiency of entrainment necessitating more cloud to produce a given entrainment rate. The seventh one is increased surface wind speed, which allows stronger shear mixing and increases latent heat flux and cloud.



Aerosol is the other controlling factor. Fan et al. (2016) reviewed aerosol-cloud interactions and found consistent conclusions
from recent studies that adding cloud condensation nuclei to warm clouds with very low drop number concentration can
invigorate them, leading to taller clouds, larger cloud water content, and higher rain rates. In trade-wind cumulus over the
Indian Ocean, a complex interaction between trade-wind cumuli and absorbing aerosols was observed using satellite data.
The cloud fraction increases when the aerosol optical depth (AOD) is smaller than 0.3, plateaus when AOD is between 0.3 –
0.45, and decreases when AOD is greater than 0.45. The cloud top height is an insensitive response to AOD. Cloud size
distribution shifts significantly to larger cloud size where AOD is greater than 0.45 (Dey et al., 2011). Mieslinger et al.
(2019) used more than 1000 Advanced Spaceborne Thermal Emission and Reflection Radiometer (ASTER) images to
investigate the shallow cumulus macrophysical properties on large-scale meteorology. They found that the surface wind
speed is the strongest control factor. Their result supported the deepening response of a wind-driven marine boundary layer
as modelled by large eddy simulations.


For mesoscale organization of precipitating shallow cumulus, Seifert and Heus (2013) simulated the formation of cold pools
by cooling due to evaporation of raindrops and found precipitation was required for aggregation. Recently, Bretherton and
Blossey (2017) examined the self-aggregation of shallow cumulus. They showed that precipitation, mesoscale feedbacks of
radiative and surface fluxes are desirable but not essential in self-aggregation. They suggested self-aggregation is more likely
in mesoscale regions of higher water vapor path. Recent numerical simulations have studied the flower cloud system on 2
February 2020 near Barbados. Narenpitak et al. (2021) examined the transition of trade cumulus organization from sugar to
flower using a Lagrangian large eddy simulation following the trajectory of the transition. They emphasised the
strengthening effect of the mesoscale total water convergence surrounding the cloud system and the accelerating organisation
by the large-scale upward motion through the buoyant turbulence kinetic energy production. Dauhut et al. (2023) also
simulated the case to investigate the convective processes. They found the organised updraughts in an arc on the leading
western edge only of the flower cloud with the highest rain rates and the largest cloud liquid water content. In spite of the
general knowledge of long history on shallow cumulus clouds, there are, however, still many unknowns with respect to, for
example, the formation of the mesoscale organization, their development and dissipation, and the effects of microphysical
processes in their life cycle. Dauhut et al. (2023) have suggested an analogy between flower cloud systems and MCSs, which
supported the idea to decompose the life cycle of the flower as previous studies have usually decomposed the life cycle of
deeper convective systems (e.g., Machado et al., 1998; Houze, 2004, 2018; Roca et al., 2017; Houze, 2018). Such studies
usually consist of three steps: the identification of a cloud system at a given time, tracking it with time, and the analysis of its
features (e.g., Fiolleau and Roca, 2013). We follow their method to characterize the flower system on 2 February 2020.

The field campaign of EUREC[4]A took place in January and February 2020 in the Trade-wind Alley of the North Atlantic
(Stevens et al., 2021), accomplished through a number of international projects, including EUREC[4]A-UK, a UK programme

of observational and modelling research which aims to study the detailed aerosol and cloud processes in the life cycle of trade cumulus clouds and the two-way interactions between the cloud processes and the large-scale dynamics.

Although there have been studies on the statistical features of the cloud organisations (e.g., Stevens et al., 2019, Bony et al., 2020), the life cycle of an individual flower system has not been examined. In this paper, we present the study of the life cycle of a flower cloud system using satellite data and investigate favourable and unfavourable conditions for this flower system. The main part of the paper is organised according to the different stages of the flower cloud system. Section 2 describes the datasets used in this paper. Section 3 shows the large-scale synoptic patterns associated with the flower.

Sections 4 and 5 presents the satellite and dropsonde analyses of its life cycle. Section 6 discusses the controlling factors. The final section presents the conclusions.

## 2 Datasets

The Geostationary Operational Environmental Satellite-16 Series (GOES-16) supplies high-resolution imagery in 16 spectral bands using its Advanced Baseline Imager (ABI). The ABI products include radiances, cloud top height, cloud optical depth,

cloud particle size (effective radius), aerosol detection and aerosol optical depth, among others (Schmit et al., 2017; Heidinger et al., 2020).

Single reflective band ABI Level 1b radiance products are used to map outgoing radiance values at the top of the atmosphere for visible, near-infrared, and infrared bands. In this paper, we use a visible band radiance (band 6 at 2.25 μm) and an

infrared band (band 14 at 11.2 μm) to identify cloudy and cloud-free regions and to view cloud top features in daytime and nighttime.

Cloud top height (CTH) is retrieved with an algorithm using ABI infrared bands. The cloud optical depth (COD) and cloud particle size distribution (CPSD) are produced by the same algorithm. COD is the measure of the extinction due to drops or

ice particles at a wavelength of 0.64 um (Band 2). The cloud effective particle size is computed using both the visible and near-infrared ABI bands during the day and the infrared bands during the night.

The aerosol detection product uses known spectral absorption and scattering properties of different aerosols at several spectral bands to detect the presence of dust or smoke aerosol in the atmosphere. The aerosol optical depth (AOD) product

measures the reflectance properties of cloud-free pixels at the top of the atmosphere (TOA) at several spectral wavelengths of the ABI.

The Moderate Resolution Imaging Spectroradiometer (MODIS) aerosol optical depth (AOD) product is retrieved using three algorithms: the Enhanced Deep Blue (DB) and Dark Target (DT) algorithms over land, and a DT over-water algorithm
(Sayer et al., 2014).

ERA5 is the fifth generation of the European Centre for Medium-Range Weather Forecasts (ECMWF) reanalysis for the global climate and weather for the past decades. It combines observations from across the world into global estimates using advanced modelling and data assimilation systems. (Hersbach et al., 2018). For a latest overview of ERA5, including the
uncertainty, please see Hersbach et al. (2020). ERA5 hourly data are used in the analysis.

Dropsondes are useful to measure the vertical profiles of atmospheric parameters and to derive the areal-mean mass divergence from a series of dropsondes along a circular pattern over oceans during airborne field campaigns (e.g., Bony and Stevens, 2019). The same sampling strategy was adopted in the EUREC[4]A campaign (Konow et al., 2021).

The French SAFIRE ATR aircraft flew mainly in the sub-cloud layer and near the cloud base level to carry out cloud microphysics, aerosol and many other atmospheric measurements. The observations, including the flight pattern and instrumentation, from the ATR aircraft during the field campaign can be found in Bony et al. (2022).

**3 Synoptic pattern**

Figure 1 shows the MODIS satellite image superimposed with the surface pressure chart on 2 February 2020. The centre of a cyclone was located at 43 °N and 62 °W. A frontal system ran from 50 N and 10 W to the Greater Antilles. Ahead of the frontal system was a high-pressure ridge that extended from the west coast of Africa towards the southwest with nearly clear sky. There was a region of stratocumulus clouds between 40 – 30 °W and 10 – 20 °N which broke up towards the west to become small cumulus clouds, as is common in this type of situation (e.g., Norris, 1998). Further to the west, the clouds
were organised in lines that run approximately northeast to southwest. So-called flower clouds (Bony et al. 2020) were evident around 60 °W, near Barbados.

**4 Life cycle of the flower**

**4.1 General features and categorization**

The detection of a cloud system is carried out by specifying a brightness temperature threshold on an IR image to mark the
border of contiguous cloudy regions where the brightness temperature smaller than the threshold (e.g., Maddox, 1980). The threshold varies from case to case, in the range of 205 K to 233 K for mesoscale convective systems with deep convections (Maddox, 1980; Machado et al., 1998, Fiolleau and Roca, 2013). After a series of tests and trials, we found that a radiance





threshold of 102 mW/(m$^2$ sr cm$^{-1}$) of GOES-16 IR channel 14 (11.2 µm) was a good choice to identify the flower system. Figure 2 shows a time sequence of GOES-16 imagery of the infrared 11.2 µm band, with the threshold being represented as red curves, every three hours from 0300 to 1800 UTC.

The cloud system was manually tracked with a time-step of 10 min from 0000 UTC to 2350 UTC. The total cloud area enclosed by the red curves was calculated at each time step to form a time series which is shown in Figure 3a. The mean CTH of the cloud system and the frequency of occurrence of the CTH are shown in Figure 3b and Figure 4, respectively,

The life cycle of a mesoscale convective system can be divided into different stages based mainly on the area, and sometimes on other factors, such as precipitation (McAnelly and Cotton, 1992). A popular categorization consists of three stages: developing, mature, and dissipating (Houze, 2004). Other studies used modified stages, for example, four stages (convective initiation, genesis, mature, and decay) by Coniglio et al. (2010), and five stages (two growing stages (40% and 80% of its maximum area), a mature stage, and two dissipating stage (80% and 40% of its maximum area)) by Wall et al. (2018). The life cycle of the flower cloud system in this paper was subjectively divided into three stages, i.e., developing, mature, and decaying stages, based mainly on the area of the system.

The developing stage occurred before 1220 UTC when the area increased with time (Figure 3a) and the majority of CTH was below ~ 1500 m (Figure 4) with some oscillations in the mean CTH (Figure 3b). The mature stage lasted from 1220 UTC to 1900 UTC when the area curve showed smaller variation over time, and the frequency of occurrence in CTH was relatively evenly distributed with a small portion (~ 6 %) exceeding 3000 m, the height between the averaged inversion top (~2900 m) and highest inversion top (~3100 m) determined by the HALO dropsondes. The decaying stage was marked by a decreasing cloud area from 1900 ITC (Figure 3a) although some clouds still remained at high levels (Figure 4).

**4.2 Developing stage**

Figure 5 displays a sequence of satellite images from 00 to 11 UTC, 2 February 2020. In fact, the cloud system developed from two cloud regions, indicated by red arrows in the top-left panel at 00 UTC (Figure 5a); they are linked by a line of clouds. The cloud-top heights at this time are estimated to be between 800 – 1500 m. The cloud region had increased an hour later (Figure 5b). The linear features had become more pronounced and the cloud region at the westerly end of the short E-W line had increased in size. There were filaments surrounding the linear cloud features. The cloud system further developed while retaining its approximate shape as it moved towards the west until 09 UTC (Figure 5c-5j) when the system lost its L-shape. The area of cloud system increased from ~ 1000 km$^2$ between 04 – 05 UTC to ~ 2000 km$^2$ between 09 – 10 UTC (Figure 3a and Figures 5e-k). At the same time, the mean CTH increased steadily before 0500 UTC and between 0900 – 1100 UTC, with peaks appearing at 0840 UTC and just before 1100 UTC (Figure 3b). The local maxima in CTH (Figure 6h – 6j, and 6l) were related to convective activity. For example, detrained cloudy air due to cumulus development can been



seen in Figures 5h – 5i. The area did not increase proportionally with the rising CTH between 0700 – 0900 UTC, which means that the development before 1100 UTC was caused by the expansion of growing cumulus.

Figure 7 displays the satellite imagery every 20 minutes from 0720 UTC to 1200 UTC, in order to examine the developing stage in more detail. Two main cloud clusters are shown in Figure 7a linked by a linear cloud as mentioned above. There was not a continuous increase in the area. A cloud region separated from the main cloud between 0720 – 0800 UTC Figures 7a-c). The area of the detrainment cloud region in the central cluster, as indicated by an arrow in Figure 7f, was seen to increase further in the next two hours. The region first became prominent at 0840 at 12.5 °N, 56.7 °W (Fig. 7e). The area of the east cluster also grew, and one cloud region in the cluster became dominant. Eventually, the two cloud regions merged at about 1100 UTC. Quite when the cloud system could be termed a flower is debatable.

A rapid increase in area occurred from 1030 UTC to 1220 UTC; the area more than doubled in less than two hours (Figure 3a). The increase in size can also be seen in Figure 8.a1-a5. The clouds developed to higher levels and were not dominated by those with top height lower than 1600 m after the fast development (Figure 4). The fast increase in cloud area between 1100 – 1220 UTC can also be seen in Figures 8.a1-a5. Some clouds developed to higher levels (Figures 8.b1-b5).

The cloud system changed from a region with scattered individual cumulus clouds with maximum values of cloud optical depth (COD) to larger regions with what looked like aggregates of clouds with COD > 120 occupying a large portion in the inner part of the system. The maximum value of COD was about 158 at 1220 UTC (Fig 8.c1-c5). The cumulus clouds are not visible in Figures 7o and 8.a1-a5, for example, due to the detrainment layer. However, it is reasonable to conclude that the cloud system consists of a multitude of individual cumulus clouds as well as aggregates of these clouds (Figures. 8c1-c5 for example). The COD is proportional to the liquid water content (LWC) along the path, but inversely proportional to the effective radius in a cumulus column (Brenguier et al., 2011; Wolf et al., 2019). Since the CTH and the cloud particle size in the high COD region (brown colour) did not change much (Figure 8.b1-b5 & d1-d5), the large values were related to high values of LWC. The maximum value of LWC is often found just below cloud top in a growing trade-wind cumulus. Several regions with a derived effective radius, $r_e$ > 20 μm can be seen in the developing stage (Figure 8.d1-d5), strongly suggesting that the warm rain processes operated in those cells. A comma-shaped area with effective radius exceeding 30 μm developed from about 1300 UTC in the mature stage (Figure 8.d7).

### 4.3 Mature stage

Three features were observed during the mature stage between 1220 – 1900 UTC.

The first feature was a large area of the CTH higher than 2500 m, which was different from earlier times when only a patch or a few local maxima in CTH appeared (e.g., Figure 9.b1-b4). The cloud-top height of a small part of the cloud reached



above 2500 m at 1220 UTC (Figure 9.b5). The area of CTH exceeding 2500 m continued to increase at 1300 UTC (Figure 9.b5). The corresponding average values of the base and top heights of the inversion determined from HALO dropsondes

on 2 February, were ~ 2600 and 2900 m, respectively. However, there was significant local variability and indeed inversion tops determined from some dropsondes reached about 3100 m. The dropsonde data at a later time (~ 1900 UTC) indicated that the inversion base height (and the maximum cloud top height) was ~ 2800 m (see Section 5 for details). A large portion of the cloud system reached the inversion layer in the mature stage.

The second feature was the production of large cloud drops near cloud top. A region with effective radii larger than 35 µm appeared at cloud top after 1300 UTC (Figure 8d6) and there were drops larger than about 45 µm at 1340 UTC in a comma shape (Figure 8d8). Notice that the radiance values are low in this region (Figure 8a8). The arc region of the comma contained drops larger than about 30 µm from 1300 onwards. Higher cloud top and large drops strongly suggest the production of warm rain in this region of the cloud system. The areal extent of the region increased with time until about

1600 UTC and the tail of the comma disappeared. The maximum size of the particles derived from the satellite observations decreased during this period. The French ATR aircraft (Bony et al, 2022) ascended through the western edge of this region of elevated effective radius at about 2.5 km (1510 UTC). Figure 9 of the size distribution derived from the microphysics airborne platform (PMA) observations shows the presence of small concentrations of cloud drops with sizes up to about 200 µm diameter, which verifies the satellite retrieval. Larger drops were also observed in the region at an altitude of about 1 km

as shown by the plots of mean volume diameter, MVD in Figure 10. The aircraft entered the east side of the cloud system from the north and while remaining at constant altitude, turned towards the west as shown in Figure 11, where the red, blue, green, and cyan colours represent four periods, n1 to n4. The aircraft passed through what appears to be below the region of interest (n3). Relatively low values of liquid water content, but large values of MVD, up to about 500 µm were observed. Two separated cloud regions were observed to the east of the region both with higher values of LWC. There were some large

values of MVD in n1 to the NE side of the system, but the MVD values were low in n2 on the east side (Figures 10 and 11).

It is clear from the ATR in-situ measurements during the period of about 14:55 to about 15:17 (Figure 12) that there were several individual small-scale cumulus clouds in two clusters (n1 and n2) separated by clear air. There was a large variation in the microphysical properties of these clouds. The cumulus clouds with the largest values of vertical velocity and liquid

water content (n1 and n2) were on the eastern side of the cloud system at this time (Figure 12). The observations perhaps suggest a different picture than the one of a classical mesoscale convective system described by the LES (Large Eddy Simulation) modelling study of Dauhut et al (2023) where the small-scale cumulus clouds were found on the western side of the cloud system only.

The third feature was the arc clouds presumably associated with one or more cold pools. The rain that likely occurred in the large-drop region discussed above would cause a cold pool and associated gust front with clouds along it if there was





evaporation below cloud base. Figure 13 shows the time sequence of satellite radiance imagery in a region between 60 °W – 57 °W and 12 °N – 14 °N from 1400 UTC to 2100 UTC. The gust fronts at the head of the cold pool appeared before 1500 UTC when cloud arcs became visible for the first time just west of 57.5 °W at 1430 UTC (Figure 13b). The cold pool

enclosed by cloud arcs expanded thereafter. Almost certainly the cold pool was the result of the area of precipitation shown in Figures 8d5-9. It is the only sequence of cloud arcs clearly associated with a cold pool observed in the satellite imagery. It is not clear if some cold pools existed earlier and were hidden by the upper-level cloud. There was only one region of precipitation seen in the satellite observations (Figure 7d9 for example). The Dauhut et al. simulations suggest the presence of many cold pools that aggregate to produce one large cold pool.

During the propagation, some cloud arcs passed through existing clouds and interacted with them. The interaction led to enhancement in cloud radiance. Such enhancement can be seen in small clouds between the flower and a large cloud region for example to the east of the flower at 1630 UTC (Figure 13f).

The propagation speeds of cloud arcs were not uniform by visually inspecting the movement of the arcs from the sequence of

imagery (Figure 13) since either the cloud arcs were loosely linked as a non-circular ring, or they may well have resulted from different rainshafts. To estimate the propagation speed alone, we used a coordinate system moving with the flower. Since not all cloud arcs were well defined in shape, we only selected two clear arcs (as indicated by arrows in Figure 13j) to calculate their propagation. We named the two arcs as the north arc and the south arc, respectively. The flower moved towards the north-west. The north arc was at the leading edge, while the south arc at the trailing edge. At the leading edge,

the propagation speed between 1800 and 2000 UTC was approximately 4 m s$^{-1}$. At the trailing edge, the speed between 1710 and 1910 UTC was about 6 m s$^{-1}$. We will discuss the propagation in Section 5.

Figures 14 – 16 show the imagery of the CTH, COD, and cloud particle size between 1400 UTC – 2100 UTC, respectively. Although there were some changes in area with CTH > 2500 m during this period (Figure 14a-k), the shape of the high CTH

did not change much until 1600 UTC. The COD imagery during 1400 – 1900 UTC shows several areas of values larger than 30, with a few local maxima of about 50 appearing from time to time.

The cloud particle size imagery shows that the area with effective radius, $r_e$ > 30 μm increased from 1400 UTC to 1600 UTC (Figure 16a-e). The derived effective radii were largest at 1400 UTC with maximum values of about 50 μm. The values

subsequently decreased thereafter; the maximum values were about 45 μm at 1430 UTC, decreasing to 45 μm or less at 1600 UTC. This is likely a result of precipitation falling out of the cloud. Taking Figures 8 and 16 into consideration, it was reasonable to link the area of large effective radius to precipitation and the formation of the cold pools.





### 4.4 Decaying stage

The cloud area of the flower decreased from 1900 - 2100 UTC (Figures 13k-13o) although the cloud top height in the centre remained greater than 3000 m (Figures 14k-14o). The areas of high COD also decreased significantly from 1900 UTC (Figure 15k-15o). Large effective radii can still be seen just north of 13 °N between 59.1 °W and 58.2 °W at 1900 UTC (Figure 16k), but the areas of large drops also decreased (Figure 16l-16o). The total cloud area, and the cloud activities, reflected by high COD and large effective radius, also decreased in the decaying stage.

**5 Dropsonde analysis**

The HALO aircraft flew a clover pattern on 2 February (Konow et al. 2021). Those dropsondes allowed us to understand the temporal variation of the vertical stratification at certain locations. It took about 15 min for a dropsonde to fall from ~ 9 km into the boundary layer. We selected 17 dropsondes that cover the period of interest and divided them into four groups, labelled as N, E, W and S, respectively (Figure 17). The locations of the dropsondes in each group were very close when 300 they reached the ground. Figure 17 shows the locations where they landed and the launch times of the selected dropsondes. The temperature, relative humidity, and wind in each group were examined in order to understand the temporal variations. The temperature profiles at each location are displayed in Figure 18.

The trade wind inversion base heights were generally higher at later times for most dropsonde. The inversion strengths, 305 defined as the temperature difference between the inversion top and inversion base, were weaker at later times. An exception was W4, where the inversion base height increased from about 2500 m in W1 to about 2800 m in W4, but the inversion top height only slightly increased from about 2800 m to about 2900 m. The inversion strength increased from 1.4 °C (13.0 °C – 11.6 °C) in W1 to 4.6 °C (13.4 °C – 8.8 °C) in W4. To investigate the reason behind the different behaviour of W4, further analysis of W1 – W4 was made. Figures 19 and 20 show the locations of those dropsondes superimposed on the visible 310 satellite image and the profiles of relative humidity, equivalent potential temperature, wind speed and direction, respectively. The four dropsondes fell in different environments: W1 in clear air far away from the flower system studied here at 1339; W2 just north of the system at 1654, W3 behind the cloud arc associated with the cold pool about an hour later at 1750, and W4 into the cloud system and behind the gust front about an hour after W3 at 1845.

Figure 20a indicates that the cloud top measured by W4 was approximately 2790 m, coincidental with the inversion base (Figure 18c). The temperatures were about 0.55 °C higher in W4 than those in W1 above the inversion top between 2950 m and 3100 m but about 1.4 °C lower below the inversion base from 1700 m to 2500 m (Figure 18c). Possible explanations for the observed profiles will be discussed in Section 6.1.2.





The thermodynamic profiles should be different if a dropsonde fell into a cold pool (e.g., Cui et al., 2014; Touzé-Peiffer et al. 2022). The variables measured by W3 indeed exhibited the characteristics of a cold pool. The mean temperatures below 200 m of W3 and W2 were 24.3 °C and 25.8 °C, respectively, approximately 1.5 °C lower at W3 location in the cold pool than at W2 just outside the cloud system (Figure 18c). The relative humidity did not decrease with decreasing height in the lowest 200 m in W3 as it did for W2 (Figure 20a). The mean wind speed below 200 m was about 2 m s$^{-1}$ greater in W3 than in W2.

The mean wind direction below 200 m was 134 degrees for W3 and 108 degrees for W2, a veering of 26 degrees.

Hutson et al. (2019) showed that the speed of a cold pool is given by c=k(gh($\rho_1$-$\rho_0$)/$\rho_0$)$^{1/2}$, where $g$ is the acceleration due to gravity, $h$ is the depth of the cold pool, $\rho_0$ is the air density of the environment, $\rho_1$ is the air density within the cold pool, and $k$ is a variable between 0.7 and 1.3 that attributes to finite channel depth effects and viscous dissipation (Howard, 2013;

Hutson et al., 2019). Using the profiles of W3 and W2, the estimated speed was calculated to be between 1.8 and 3.4 m s$^{-1}$. The larger speed of 3.4 m s$^{-1}$ is slightly smaller than the 4 m s$^{-1}$ speed estimated with the satellite data as discussed in the above section. The smaller propagation speed could be caused by the fact that the location of W3 was not just behind the gust front, but further way, or the value of $k$, or the environmental wind shear (Liu and Moncrieff, 1996).

## 6 Environmental conditions

We now examine the atmospheric conditions in the environment of the cloud system as it travels from the NE towards the EUREC[4]A circle in order to explain the changing cloud characteristics and formation of the flower.

### 6.1 Atmosphere-ocean interface

Three parameters at the atmosphere-ocean interface influence cumulus clouds: sea-surface temperature (Klein et al., 2017), the advection of air temperature (Klein et al., 2017), and the wind speed near the surface (Nuijens and Stevens, 2012; Klein

et al., 2017; Mieslinger et al., 2019 ).

Figures 21 a and b show the longitudinal-temporal variation of sea surface temperature and wind vectors at 10 m, and wind speed at 10 m, respectively, from the ERA5 dataset. The clouds discussed in this paper were located between 54.6 –53.7 °W at 00 UTC where the SSTs (Sea Surface Temperature) were ~299.9 K. The wind speed and direction were about 7.5 m s$^{-1}$ and

60 °, respectively. The cloud system moved to -56.3 – -55.3 °W at 06 UTC. The SSTs increased to ~300.0 K, and the wind speeds increased to ~8.9 m s$^{-1}$ and the directions veered to ~71 °. It moved further toward west to between -57.9 – -56.9 °W at 12 UTC. The SST increased to ~300.2 K, but the wind speeds decreased to ~6.6 m s$^{-1}$ and the directions from east at ~90 °. It moved to -59.2 – -58.3 °W at 18 UTC, and the SSTs changed to ~300.1 K. The wind speeds further decreased to ~5.1 m s$^{-1}$, and the directions were ~97 °.






The increase in SSTs plays an important role in increasing the boundary layer height, which allows for higher cloud top (Karlsson et al., 2010). Cold air advection into a region near the surface causes an increase in the upward buoyancy flux and therefore can promote the formation of more clouds depending of course on the thermodynamics in the cloud layer (Norris and Lacobellies, 2005). Figure 21a indicates clear cold advection at the surface along the path of the cloud system. Those

two factors favour more and higher clouds. Vial et al. (2019) also found a westward deepening of clouds as SSTs increased westward in their study of the daily cycle of trade-wind cumulus.

An increase in the surface wind speed promotes surface-driven shear mixing and latent heat flux, and, thus, can boost the production of more clouds (Brueck et al., 2015). Nuijens et al. (2009) and Nuijens and Stevens (2012) found that high wind

speeds near the surface result in a deepening of the cloud layer and increased area rainfall due to an increase in boundary layer humidity. Figure 21b shows that the surface wind speeds were ~ 8 – 9 m s$^{-1}$ along the path of the cloud system from ~54 °W to ~57.5 °W from 00 UTC to 10 UTC. However, the speeds decreased from ~7 m s$^{-1}$ at 10 UTC to ~5.5 m s$^{-1}$ between 60 °W and 55 °W, which disfavoured the flower development.

**6.2 The inversion layer**

Temperature profiles measured with the dropsondes revealed the structure and strength of the inversion layer in Section 5. The inversion layer of W4, capping the cloud system, became stronger from ~ 1300 to ~1800 UTC and did not rise as at other locations during the similar period. To understand the difference between the W4 inversion and W1- W3, we used the ERA5 data to find the temperature change above and below the inversion layer. Figure 22 shows the longitudinal-temporal variation of temperature and wind vectors at 700 hPa and 750 hPa, as well as the temperature difference between the two

levels. we checked whether the reanalysis data was close to the observations. The corresponding temperatures of ERA5 at the dropsonde times and locations were close to the temperatures measured with the dropsondes, and most of the differences were between 0.1 – 0.6 °C. The temperatures and differences at those times and locations in ERA5 data agree well with the dropsonde data.

Both the dropsonde and the reanalysis data indicate changes in temperature at 700 hPa and at 750 hPa with time and location

along the trajectory. The temperature drop can be seen clearly from 0600 UTC to 1900 UTC between 60 °W – 50 °W in Figures 22a-b. The temperature decreased slightly more at 750 hPa than at 700 hPa between 1600 – 1800 UTC (Figure 22c), which was consistent with the dropsonde temperature profiles in Figure 18c which shows the decrease below the inversion base in W4. The slight warming above the inversion top in W4 could be induced by the mesoscale subsidence in the overlying inversion (Bretherton and Blossey, 2017). Observational studies of cumulus clouds during the Barbados Aerosol

Cloud Experiment (Jung and Albrecht, 2014) revealed that the air above the cloud top descends along the downshear side of the cloud edge. The somewhat larger cooling near the inversion base could be caused by evaporative cooling of detrained cloudy air just above cloud top, and/or the radiative cooling due to the reflection of radiation back to the space by cloud top.



The stronger inversion in W4 seemed likely to be the result of the mesoscale temperature change and the cloud radiative effect.

The temporal change of the lapse rate of temperature depends on the differential horizontal advection of temperature, the vertical lapse rate advection, the stretching effect, i.e., $\partial w/\partial z \times (\Gamma_d - \gamma)$, where $\Gamma_d$ and $\gamma$ are the the dry adiabatic lapse rate and the lapse rate, and the differential diabatic heating (Markowski and Richardson, 2010). Figure 23 shows the longitudinal-temporal variation of vertical velocity in pressure coordinates. A region of the vertical velocity smaller than -0.1 Pa/s (relatively stronger upward motion) was located between 58.3 °W – 51.5 °W from 1200 UTC – 1700 UTC at 700 hPa. A 390 larger area of the vertical velocity less than -0.1 Pa/s at 750 hPa was located in the similar region as at 700 hPa. Stronger upward motion, through the vertical lapse rate advection, resulted in the rising inversion layers with time at other locations except in the west. Reduced subsidence increases the MBL (Marine Boundary Layer) height and allows higher clouds (Myers and Norris, 2013). The weakening of the inversion layer was probably caused by the differential horizontal advection of temperature since the temperature gradients to the east of 58 °W were weak at 700 hPa, but strong at 750 hPa.

The dropsonde analysis indicates that the cloud top was capped by a strong inversion (W4 in Figure 18c). A strengthened inversion reduces mixing across the inversion layer and keeps the boundary layer more humid and cloudier (Wood and Bretherton, 2006).

## 6.3 The boundary layer

Figure 24 exhibits the ERA5 relative humidity data at 700 hPa, 750 hPa and 850 hPa, respectively. The values of relative 400 humidity at 700 hPa were very low, rarely exceeding 20% west of 45 °W on the day, in agreement of the measurements with the dropsondes (Figure 20a). The moisture just above the inversion in the free troposphere also affects clouds. Air with higher humidity leads to reduced entrainment drying and can result in more clouds in the MBL (van der Dussen et al., 2015). The dropsonde data indicated that the cloud top height and pressure at ~1900 UTC were ~2790 m (Figure 20a). The altitude of 750 hPa was ~ 2580 m, below cloud top level although some convective cells penetrated above the inversion layer (e.g., 405 Figure 14). The relative humidity at 750 hPa was ~ 20% at 0000 UTC, decreased to ~ 10% at 0600 UTC, then increased steadily to ~ 28% at 1800 UTC along the path of the cloud system. A region of 30% – 40% relative humidity was located from -60 °W to -58 °W between 1500 UTC and 2100 UTC. At 850 hPa in the middle of the boundary layer, the relative humidity along the cloud system between 0000 UTC and 0800 UTC was ~ 30% – 40%. The relative humidity increased to ~ 55% – 80% after 0800 UTC along the path of the cloud system. The cloud system was in a moist environment in the 410 boundary layer during the developing and decaying stages.

Figure 25 shows the total precipitable water (TPW), a GOES-16 Advanced Baseline Imager product, which is computed from the retrieved atmospheric moisture profiles and represents the total integrated moisture in the atmospheric column from





the surface to the top of the atmosphere, between 0600 – 2000 UTC. The area with TPW > 26 mm occupies roughly a third

of the western region at 0600 UTC (Figure 25a). A cloud (indicating by a red arrow) is seen as a small grey area in the north-west of a belt of relatively high TPW along 12.5 N between 56.5 – 54.5 W. The higher TWP area associated with the cloud moved westerly and joined the large area of higher TPW from 1000 UTC. The area of the cloud system increased visibly up to 1300 UTC. The higher TPW surrounding the cloud system from 0600 UTC – 2000 UTC indicates higher water vapour content, which is consistent with Figure 24c. The TPW surrounding the cloud system was still high even after 1900 UTC

(Figures 25.o-p), which suggested that moisture supply was not a reason for the decaying of the system.

Bretherton and Blossey (2017) found that mesoscale cumulus aggregation likely takes place in mesoscale regions of higher water vapor path. Narenpitak et al. (2021) found that the total water path increases in the moist regions where clouds aggregate, due to or in association with mesoscale circulation rendering a net convergence of total water in the already moist

and cloudy regions, as a result, strengthening the organization. Our analyses of relative humidity and TPW lend observational support to their conclusions.

Figure 26a shows that the AOD increased from north-east towards the region where the flower was located. The aerosol detection product indicated that there was no noticeable dust or smoke aerosol in the Barbados region. The MODIS

(Moderate Resolution Imaging Spectroradiometer) also detected higher AOD in the Barbados region (Figure 26b), similar to the GOES-16 AOD. Previous studies have shown that the aerosol size distribution is critical for the development of cumulus in the Caribbean region (e.g., Blyth et al., 2013). Chakraborty et al. (2016) investigated the relative influence of meteorological conditions and aerosols on the lifetime of deep mesoscale convective systems over tropical continents. They found that AOD can explain up to 24% of the total variance of the lifetime of the systems although this influence is not as

strong as that caused by convective available potential energy, relative humidity, and vertical wind shear. Although aerosol alters the lifetime of MCSs, whether or not it affects the lifetime of a flower remains unknown. Previous studies have shown that higher aerosol leads to an increase in the number of deeper cumulus clouds, but there is little increase in the frequency of higher rain rates (Spill et al., 2019). Convective clouds can also transport aerosol to higher levels (Cui and Carslaw, 2006; Chen et al., 2012). The chemical and physical properties of the aerosol particles and their origins are not clear. Ground-based

measurements of aerosol were conducted during the campaign. Instruments at the ground station were used in a previous campaign to identify the aerosol properties (e.g., Marsden et al., 2019). The sources of aerosol can be investigated using back trajectory analysis as carried out in previous campaigns (e.g., Liu et al., 2018). Further work needs to be carried out in order to examine the amount of aerosol particles that can serve as cloud condensation nuclei and their role in the invigoration of cumulus.





## 7 Conclusions

In this paper, we present an analysis of the characteristics and environmental conditions of a flower cloud system that occurred during the EUREC[4]A field campaign. The satellite data shows that the flower system started from two small linear cloud regions in one of the cloud streets to the north-east of the region. The two clouds grew into clusters during their movement towards the south-west approximately at a speed similar to the steering-level winds at 850 hPa. The area of the clusters rapidly increased over the period of about 12 hr as a result of deepening and aggregation with detrainment due to the inversion. The clusters underwent splitting and merging during the development stage. A comma-shaped area of large drops developed in the central part of the flower in the mature stage. Finally, cloud arcs appeared as a result of propagating gust fronts at the head of cold pools. The propagation speeds of the arcs were not uniform. An arc at the leading edge in the north moved at a speed similar to that of the flower, but an arc at the trailing edge in the south moved much faster due to a channelling effect.

The environmental factors in the vicinity of the developing system were considered. The SSTs along the trajectory of the flower were 26.6– 27.3 °C, slightly higher than the mean SST (26.2 °C) for flowers in Bony et al. (2020). The wind speed increased from 6.4 m s$^{-1}$ to 9 m s$^{-1}$ in the first six hours of the day and decreased late to ~5 m s$^{-1}$ in the decaying stage. The overall wind speed was weaker than the mean value for flowers of 9 m s$^{-1}$ of Bony et al. (2020), but still within the wind speed range. The inversion layer to the east of the flower system became higher and weaker with time. It was only slightly lifted but became stronger within the flower system, which favoured the horizontal spreading of detrained clouds and was consistent with the finding that a flower tends to occur with a strong inversion (Bony et al.,2020). The results indeed fall within the ballpark of climatological conditions associated with the flower pattern described by Bony et al. (2020). Our case study shows that the environmental conditions exhibited temporal variations along the trajectory of the flower system. Other conditions that favoured the cloud development included the enhanced moisture around the flower and the increasing aerosol optical depth near Barbados.

Additional future analysis will help to understand the internal structure of the flower and the microphysical processes in the life cycle of the flower. Aircraft measurement and radar observation may reveal some interesting features. For example, whether the flower system operated as a large cloud in the mature stage and whether the cloud arcs were produced by one strong downdraught in the dissipating stage. Evaluating the controlling mechanisms and processes in the development of the flower requires the use of other approaches, e.g., in-situ measurements of aerosol and cloud microphysics, and numerical modelling.

**Data availability**

Data access to GOES-R Series data and other information is available at NCEI, as well as at Comprehensive Large Array-Data Stewardship System (CLASS): https://www.avl.class.noaa.gov/saa/products/welcome.



ERA5 data is available on the Copernicus Climate Change Service (C3S) Climate Data Store: https://cds.climate.copernicus.eu/#!/search?ext=ERA5&type=dataset.

ATR's EUREC4A data are available on AERIS (cf Table 11 of Bony et al. 2022 for the complete/more specific reference). The code and data on HALO are freely available at the locations specified in Table 6 of Konow et al. (2021). .

**Author contributions**

ZC and AB prepared the manuscript with contributions from all the co-authors. ZC analysed the satellite and reanalysis data.
AG and RB contributed to the cold-pool propagation. analysis. S Bony and S Boeing made significant edits.

**Competing interests**

The contact author has declared that neither they nor their co-authors have any competing interests.

**Acknowledgements**

The authors gratefully acknowledge the efforts of the personnel associated with the British Antarctic Survey Twin-Otter operation. The pilots, Andrew Van Kints, Andrew Vidamour flew the many missions during the project. The authors acknowledge the efforts of the personnel associated with the French ATR42 and the German HALO aircraft and analysis provided by Pierre Coutris and Marie Lothon. We acknowledge the NCAS Atmospheric Measurement and Observation Facility (AMOF) a UKRI-NERC funded facility, for providing the HVPS cloud probe. We are grateful to Douglas Parker for
the inspiring and fruitful discussions of the case. We are very grateful to Bjorn Stevens who worked with Sandrine Bony in leading such an comprehensive and outstanding project. The authorswould also like to thank David Farrell, principal of the Caribbean Institute for Meteorology and Hydrology (CIMH). We would like to thank Professor Douglas Parker for the inspiring and fruitful discussions of the case. We thank National Oceanic and Atmospheric Administration (NOAA), The National Aeronautics and Space Administration (NASA), and The European Centre for Medium-Range Weather
Forecasts(ECMWF) for providing the publicly available datasets used in this study.

**Financial support**

This project was funded by the UK's Natural Environment Research Council (grant no. NE/S015868/1). Zhiqiang Cui is partly funded by the Centre for Environmental Modelling And Computation (CEMAC), University of Leeds.

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







**Figure 1: The MODIS satellite image imposed with the surface pressure chart on 2 February 2020.**




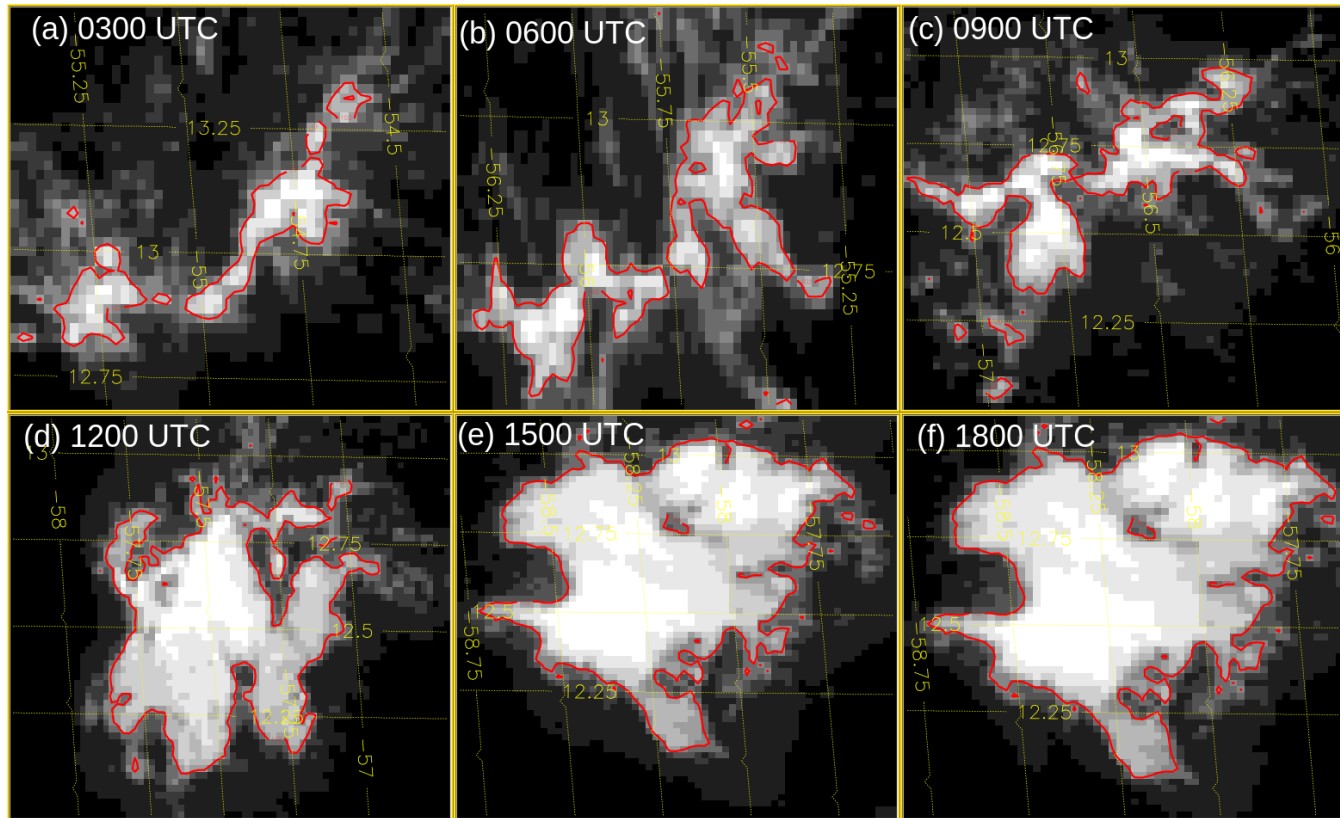

**Figure 2:** Time sequence of GOES-16 imagery of the infrared 11.2 µm band every three hour from 0300 to 1800 UTC on 2 February 2020. The red curve represents the radiance of 102 mW/(m$^2$ sr cm$^{-1}$).





**Figure 3: Temporal variation of (a) the area of the cloud system which is defined by the enclosed area with the red curve in Figure two, and (b) the mean cloud top height of the cloud system.**





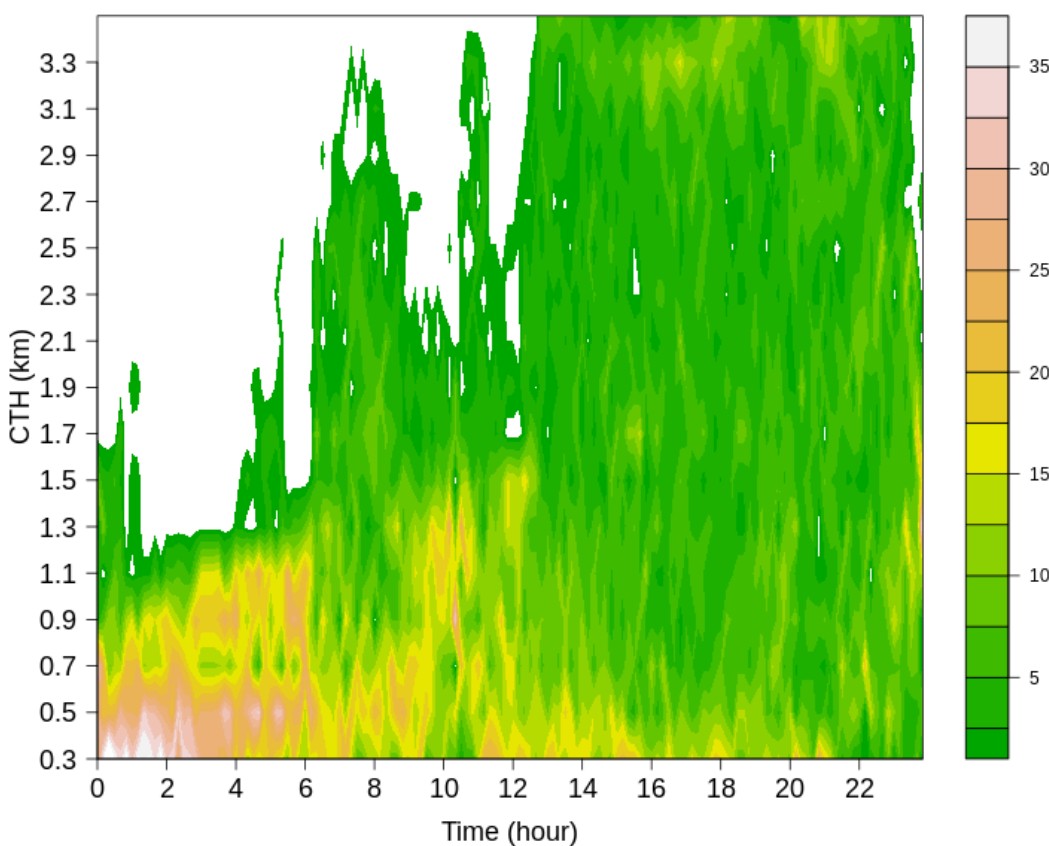

**Figure 4: The frequency of occurrence of cloud top height of the cloud system.**



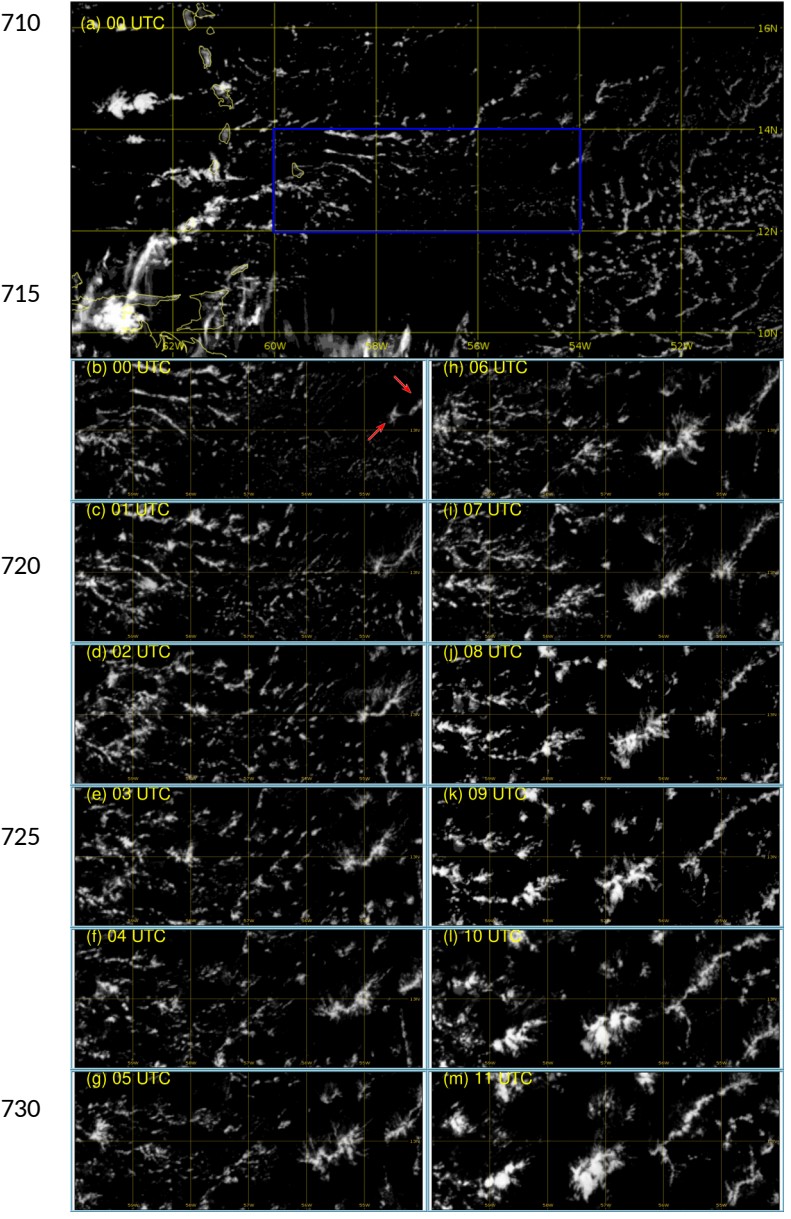

**Figure 5: Time sequence of GOES-16 infrared images between 0000 – 1100 UTC (b-m). The region is between 60 °W – 54 °W and 12 °N – 14 °N as shown by the blue box in (a).**



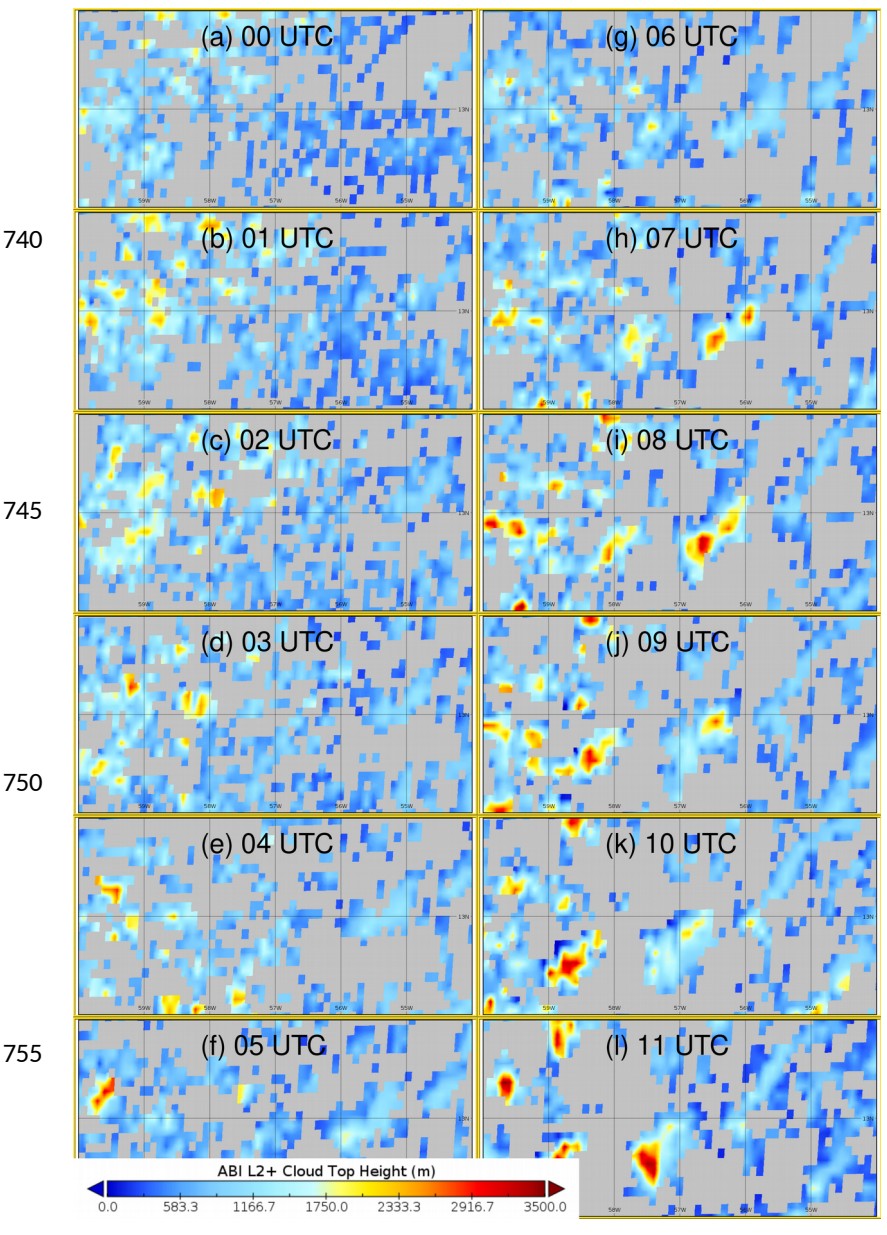

**Figure 6: Time sequence of GOES-16 cloud top height images between 0000-1100 UTC. The region is between 60 °W – 54 °W and 12 -14 °N.**







**Figure 7: Time sequence of GOES-16 infrared images between 0720 – 1200 UTC. The region is between 58 °W – 55 °W and 11.5 °N – 13.5 °N.**

765



**Figure 8: GOES-16 radiance (first column), cloud top height (the second column), cloud optical depth (the third column), and the cloud particle size (the last column) every 20 min between 1100 UTC (the first row)-1340 UTC (the last row). The region is between 59-56 °W and 11.5-13.5 °N.**



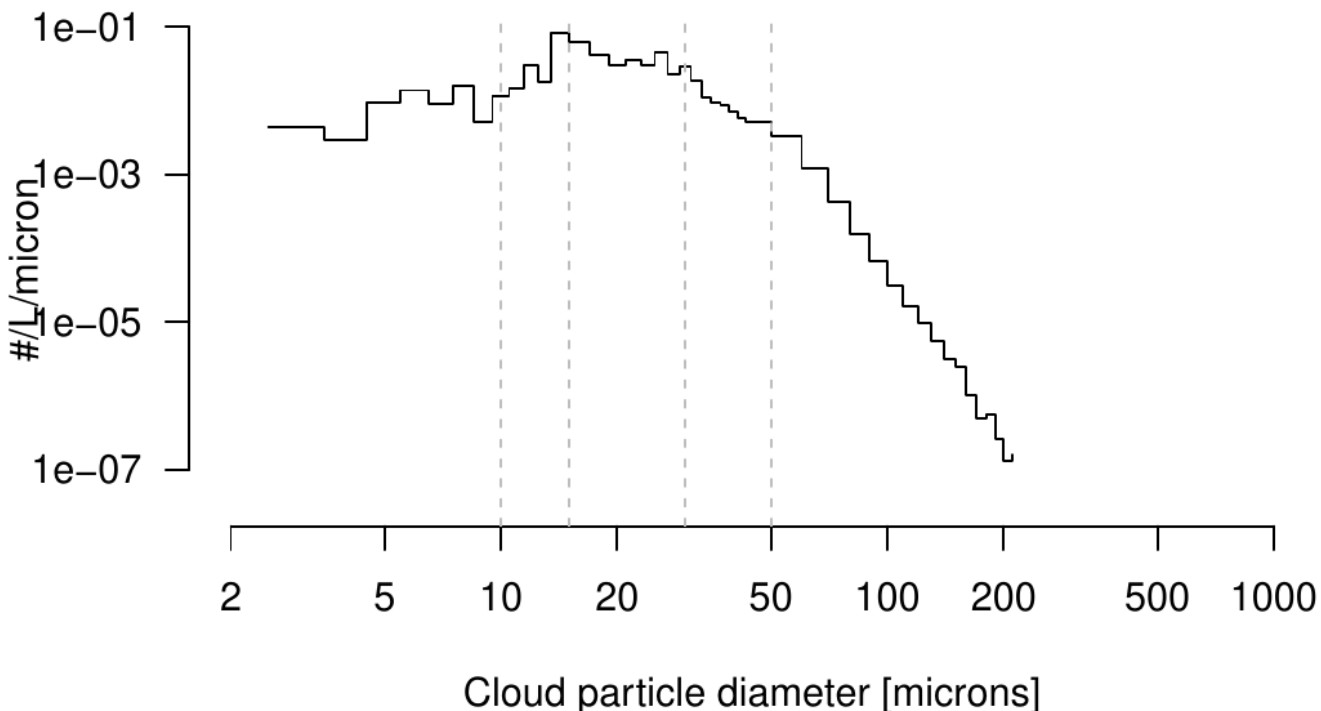

Figure 9: The averaged drop size distribution measured with the PMA microphysical data from the ATR aircraft.







**Figure 10: Median volume parameter from PMA measurements from the ATR aircraft .**



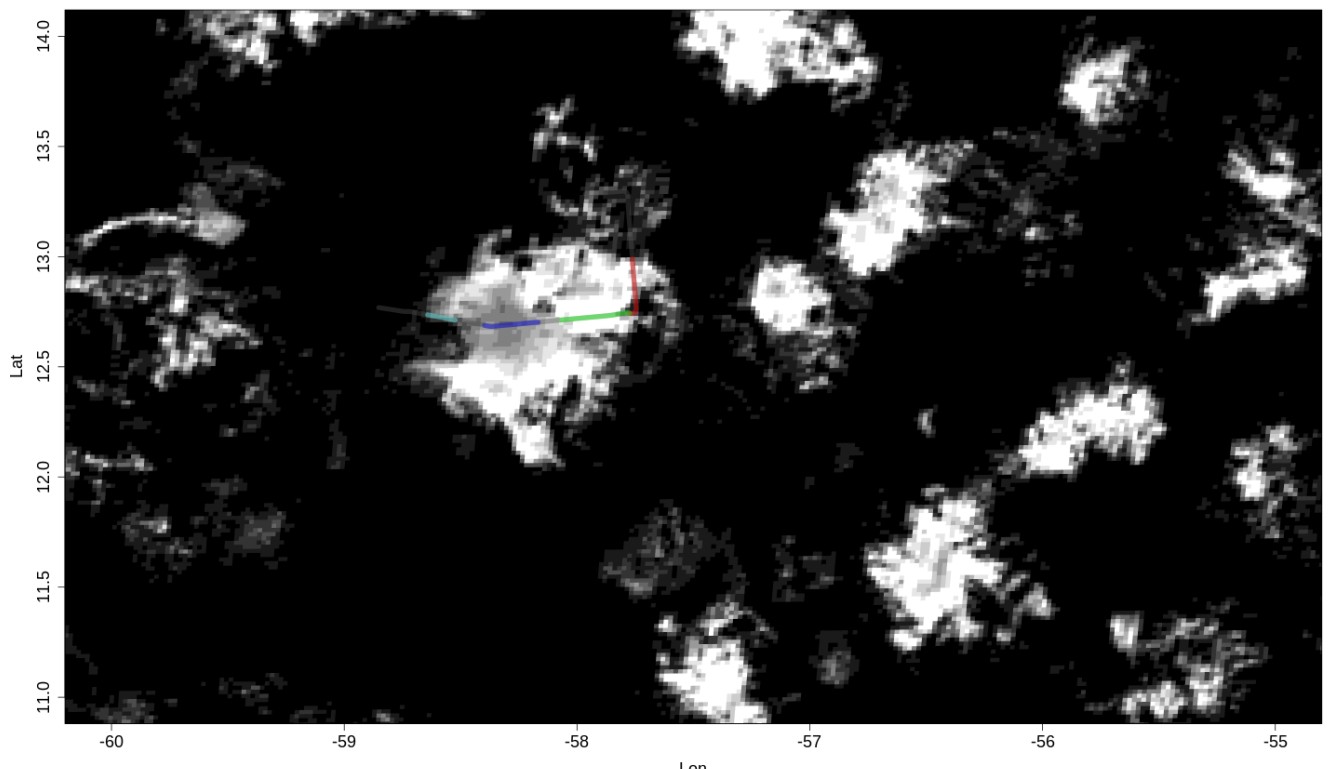

**Figure 11: GOES-16 satellite image at 1500 UTC. The ATR flight track is represented as a thick coloured curve. The red, green, blue, and cyan are four clouds measured in Figure 12.**





**Figure 12: The Vertical velocity (unit: m s⁻¹) and the LWC (unit: g m⁻³; with n1-n4 represent four sections in Figure 11) at 1 Hz from SAFIRE-CORE between 14:53:08 and 15:18:58 UTC.**





**Figure 13: GOES-16 near-IR radiance images of the 2.24 µm band between 1400 UTC – 2100 UTC. The region is between 60 °W – 57 °W and 12 °N – 14 °N.**



**Figure 14: GOES-16 images of the cloud top heigh between 1400 UTC – 2100 UTC. The region is between 60 °W – 57 °W and 12 °N – 14 °N.**

785





**Figure 15: GOES-16 images of the cloud optical depth between 1400 UTC – 2100 UTC. The region is between 60 °W – 57 °W and 12 °N – 14 °N.**





810



**Figure 16: GOES-16 images of the cloud particle size between 1400 UTC – 2100 UTC. The region is between 60 °W – 57 °W and 12 °N – 14 °N.**





**Figure 17: Location and launch time (in bracket) of dropsondes. See text for details.**



**Figure 18: Temperature profiles of the dropsondes. For location and launch time, see Figure 13.**



820

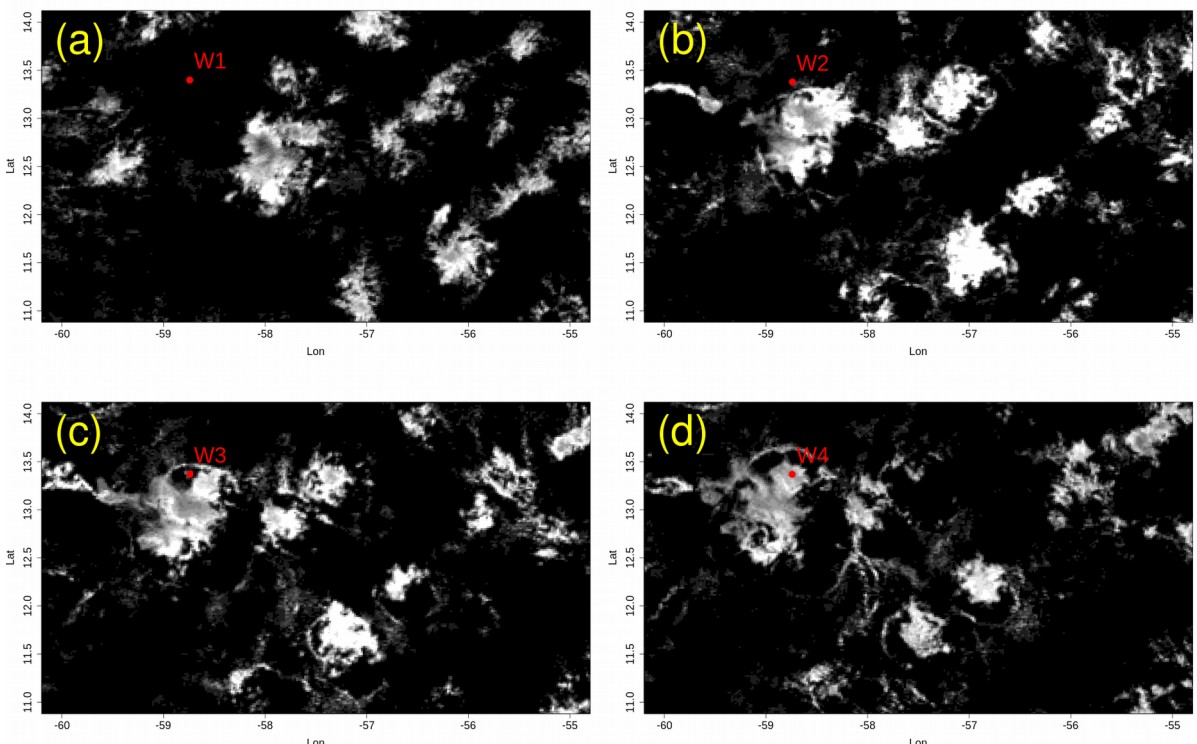

**Figure 19: Locations of the dropsondes imposed on the GOES-16 satellite images for dropsonde W1-W4.**





**Figure 20: Profiles of (a) relative humidity, (b) equivalent potential temperature, (c) wind speed, and (d) wind direction of dropsonde W1 – W4.**



**Figure 21: Longitudinal-temporal variation of (a) sea surface temperature and wind vector at 10 m, and (b) wind speed at 10 m. The black points represent the mean longitude of the cloud system. The datasets used are ERA5 at 0.25° resolutions.**



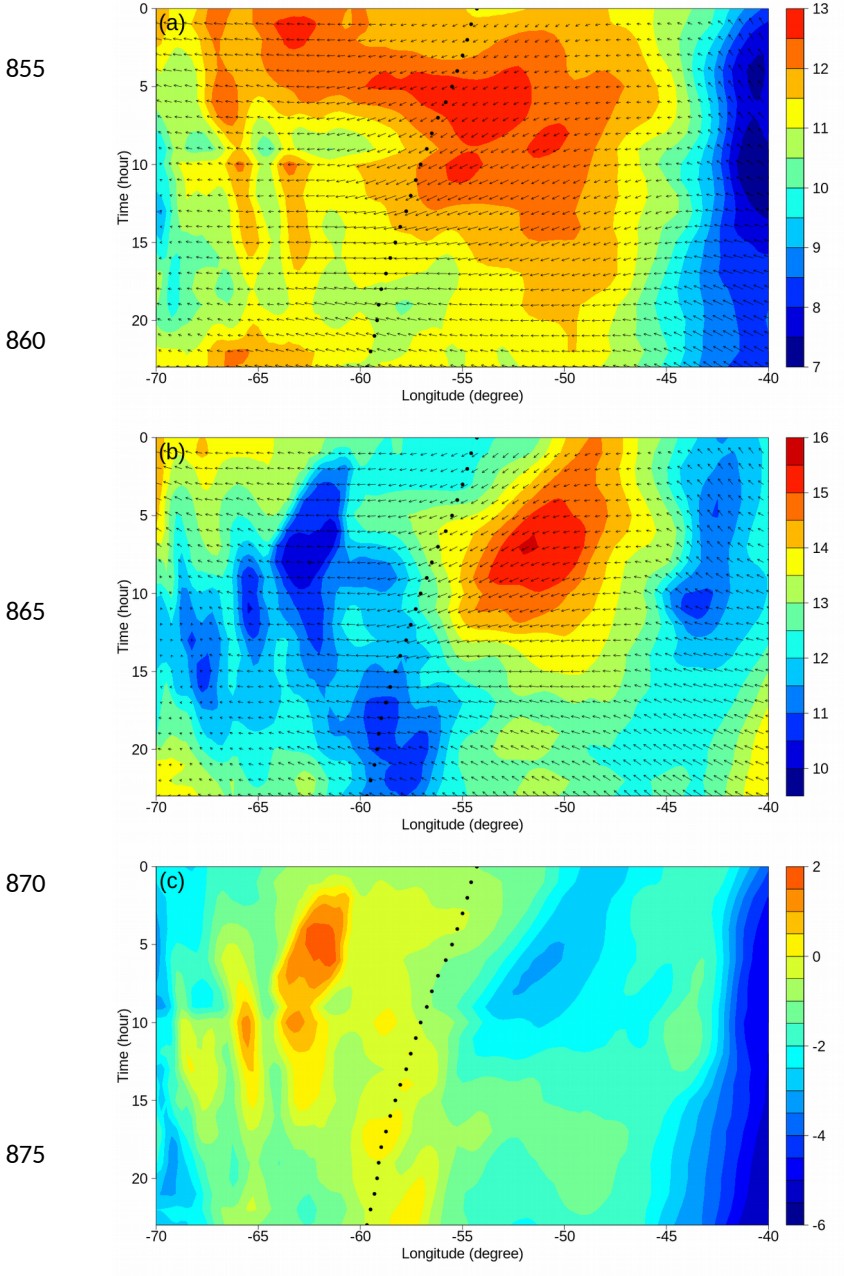

**Figure 22: Longitudinal-temporal variation of temperature and wind vector at (a) 700 hPa and (b) 750 hPa, and (c) the temperature difference between 700 hPa and 750 hPa. The black points represent the mean longitude of the cloud system. The datasets used are ERA5 at 0.25° resolutions.**



**Figure 23: Longitudinal-temporal variation of vertical velocity** in pressure **coordinates (unit: Pa/s) at (a) 700 hPa and (b) 750 hPa.
The black curves represent the vertical velocities are zero.The black points represent the mean longitude of the cloud system. The
datasets used are ERA5 at 0.25° resolutions.**




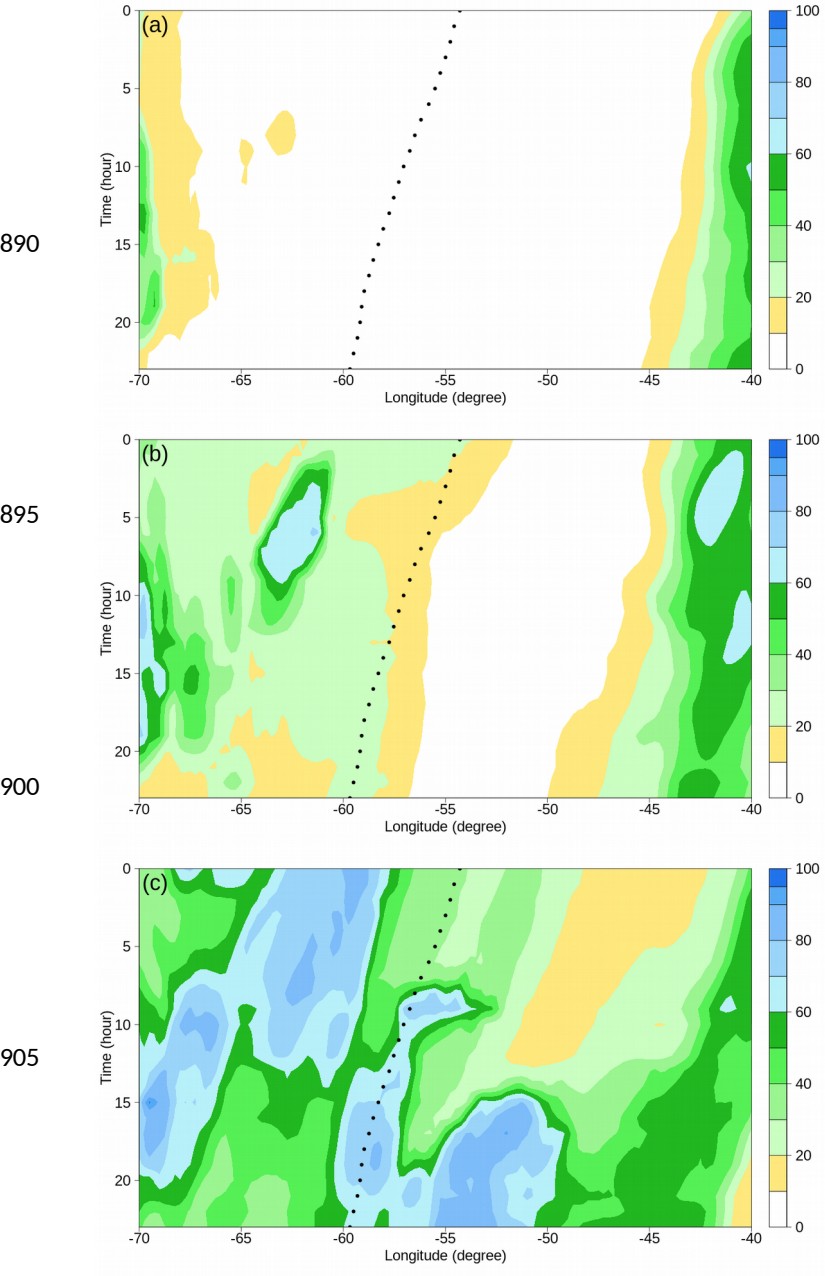

**Figure 24: Longitudinal-temporal variation of relative humidity at (a) 700 hPa, (b) 750 hPa, and (c) 850 hPa. The black points represent the mean longitude of the cloud system. The datasets used are ERA5 at 0.25° resolutions.**



**Figure 25: ABI-L2 total precipitable water between 0600 UTC and 2000 UTC.**





**Figure 26: (a) Aerosol optical depth derived from GOES-16 and (b) and from MODIS.**