# Peer review of "Life cycle of a flower cloud system during the EUREC4A field campaign"

_EGUsphere, 2023_

## Referee Comment (RC1)

**Review**

Life cycle of a flower cloud system during the EUREC4A field campaign

By Zhiqiang Cui et al.

Mesoscale cloud patterns of shallow convection are currently studied to great detail in all climate regions from polar cold-air outbreaks to green cumulus over the Amazon. In the downstream Atlantic trades, Stevens et al. (2020) observed four frequently occurring patterns: Sugar, Gravel, Flowers and Fish. This manuscript focuses on the Flowers pattern and describes based on a case-study the evolution of a cloud system within this pattern by utilizing a wide range of observations from the EUREC4A campaign.

Understanding the evolution of these mesoscale patterns is an important factor to simulate the cloud radiative feedback. As Bony et al. (2020) pointed out, each pattern is associated with a unique cloud radiative effect. Any distribution changes of these patterns that can occur due to e.g., shifts in storm-tracks or a widening of the tropics (Schulz et al., 2021), can potentially affect the radiation balance. As a consequence, these cloud patterns need to be correctly represented in both climate models for future projections and large-eddy simulations to gain a better process-understanding of their mesoscale dynamics.

The manuscript describes the life cycle of a cloud system within a Flowers pattern to great length both at the mesoscale with the help of geostationary satellite images and retrievals as well as on the cloud-scale with in-situ flight measurements and dropsonde data. While the description presented here is very elaborate, the manuscript should be more condensed and have its main audience in mind. Particularly modelers are interested to have a benchmark dataset or key parameters and their values as a reference to improve their simulations. The manuscript needs to better carve out these key parameters and be more specific in the conclusion to become impactful. That said I'm convinced that the topic of the paper will be a welcomed study to the community. As the framing needs to be improved and the list of comments is relatively long, the manuscript needs major modifications.

**General Comments**

- The manuscript would greatly improve by being more concise and more fluent. Paragraphs are often stand-alone and not well connected into the flow of the manuscript. An example is for example the 3rd and 4th paragraph in the introduction that mainly review cloud controlling factors and aerosols without being a strong part of the motivation of the manuscript.
- Please gear towards modelers and emphasize that this is observational work. The manuscript should provide clear key parameters and their values. While this is a case study and cannot be generalized, key parameters of each stage in the life-cycle shall be summarized. This boils down to the question of what added-value does this manuscript provide if someone wants to set-up and compare their simulation with observations or study their dynamics more generally.
- The manuscript would greatly benefit by being carefully read by an English native speaker (e.g. one of the co-authors).

- Plenty of studies came out of the EUREC4A field campaign and several of these touch on aspects of this study as well, in particular Feb 2. The manuscript would greatly improve if it would embed previous results to strengthen its hypotheses. Examples are the mesoscale circulations and subsidence measured by George et al. (2023), the cold pool analysis by Touze-Pfeiffer et al. (2022) and the radiative cooling profiles by Albright et al. (2021). All of these studies present observations of Feb 2$^{nd}$ and are hardly mentioned in this manuscript. Cold pool dynamics that Dauhut et al. (2023) for example mentioned are described but hardly discussed in terms of their potential role although they seem to be essential for this pattern.
- The aerosol section is very limited and less mature. I would encourage the authors to exclude any aerosol related analysis from this study.
- Precipitation, which is closely connected to cold pools and has been pointed out to correlate with Flowers has been mostly ignored. An analysis of the precipitation, e.g., from satellite products or research vessels equipped with radar (e.g. Fig 8 of Touze-Pfeiffer et al. (2022)) should be included.

**Specific comments**

- Consider capitalizing the mesoscale pattern names and make them italic for better readability
- A reference to the PMA instrument suit and its measurements is missing in the data section.
- Further, the manuscript mentions the SAFIRE-CORE dataset, which should be cited and explained in the data section as well.
- Data availability section:
  - Please provide DOIs to the specific datasets used. URLs are not a permanent identifier and, in this case, do not point to the specific versions used.
    - DOIs of GOES-16 products used
    - DOI of JOANNE/dropsonde dataset
    - DOI of SAFIRE-CORE dataset if it has been used as indicated in the manuscript
    - DOI of ERA5 hourly data product
    - Any additional datasets
  - Code to reproduce results and figures is missing
- The number of figures needs to be drastically reduced. A few suggestions are proposed in the figure specific comments, but additional deletions/merges or moves to the supplement would benefit the focus of this manuscript.
- The manuscript is studying the life cycle of a cloud system within a Flowers pattern, which according to Vial et al. (2019) undergoes a diurnal cycle. However, the analysis here uses the foundation temperature of ERA5 which is lacking a diurnal cycle. Using the skin temperature might improve some of the robustness of arguments and should be used here instead. In-situ observations from ships would be even better.
- Dropsonde analysis section:
  - Here would be a good point to reference George et al. (2021)
  - 17 dropsondes have been selected but only 4 were mentioned in the manuscript (W1 – W4). All information about grouping and soundings at N, E, S end of the circle can therefore be discarded, including Fig. 17.
  - Fig. 18 c can be combined with Fig. 20, while Fig. 18 a,b,d can be deleted

Line by line comments

- L. 55: …inversion and keeps the boundary layer….
- L. 58: please use the word cloudiness instead of cloud
- L. 64: Aerosol is an other controlling….
- L. 70ff: The work of Mieslinger et al. (2019) would better fit into the previous paragraph and should be better integrated there.
- L.86: …edge only of one Flower cloud patch with the highest…
- L. 95: The Trade wind Alley is a jargon word that seems to be better described here by "downstream trades"
- L. 101: "the life cycle of an individual flower system has not been examined". While this is true, the work of Dauhut et al. (2023) and Narenpitak et al. (2021) should definitely be mentioned here. It should also be emphasized that this study uses observations.
- L.102: The outlook to "investigate favorable and unfavorable conditions for this flower system" is not given in the manuscript. Only one cloud system is observed and neither sensitivity runs are made nor are a significant number of cloud systems observed to correlate the cloud pattern with cloud controlling factors.
- L.108: Switch "Geostationary Operational Environmental Satellite-16 (GOES-16)" with "Advanced Baseline Imager (ABI)". It's the imager which provides images and not the satellite itself.
- l. 115: please add a reason for why band 14 has been used which is not in the clean infrared window but is affected by water vapor
- l.118f: please provide a reference for the mentioned algorithms
- l.137f: The circling strategy of the EUREC4A flights has not been used here and is unimportant for this manuscript. Please leave this out and rather explain how the inversion base and inversion top are derived from the dropsonde dataset.
- L.137f: George et al. (2021) seems to be a better reference to the JOANNE dropsonde dataset. If JOANNE has not been used here, it should be justified.
- L.142: …aerosol and among other atmospheric measurements….
- L.151: The C3ONTEXT dataset of Schulz (2022) could serve as a reference here to support the presence of a Flowers pattern.
- L.158: …GOES-16 ABI IR channel… it is the IR channel of the ABI and not GOES-16. Please adapt all occurrences in the script.
- L. 159: see above
- L.158: brightness temperature would be more informative and intuitive then radiance, especially with respect to model simulations. Please convert the radiances in the script to brightness temperatures.
- L.168: "Other studies used modified…". Unimportant references as only the Houze (2004) will be used. Consider writing "We adapted this system here under the assumption that Flowers are MCSs following Dauhut et al. (2023)"
- L.171: …divided into those three stages and based mainly on the area of the system."
- L. 178: … and highest inversion top (~3100m) was determined by…
- L.179: 19 UTC

- L.196: 0720 – 0820 UTC (separation is not visible at 0800 UTC)
- L.197: Figures 7a-7d).
- L.198: …The area to the east…
- L. 199f: "the two cloud regions merged at about 11 UTC." This is vague and could be more specific. The clouds seem clustered already in 7a s there is no clear-sky separating them. Maybe one could say that by 11UTC a common stratocumulus has been developed.
- L. 215ff: Units are weirdly formatted here and throughout the manuscript. Please use the same font for $\mu$ as for other characters.
- L. 240ff: the "mean volume diameter" needs to be better defined as well as the entire PMA observations. A good place for this would be the data section.
- L. 250: where does the vertical velocity and LWC come from? Please define this in the data section.
- L. 250: at which altitude did the airplane sample these measurements? Was the flight level constant (above sea level or pressure)?
- L. 250ff: Figure 12 has been taken at 15 UTC, when the ATR has finished n2, which let me conclude that the highest values of LWC and upward motion were recorded at the WESTERN side of the cloud system, which would be in agreement with Dauhut et al. (2023).
- L. 251ff: this is too vague.
- L. 305: how are inversion top and inversion base derived from the dropsonde data? Please add this information to the data section.
- L. 351ff: What do change in boundary layer height due the authors expect from a change in sea surface temperature of 0.3K? It would be good to include here the CTH estimates from the satellite measurements and how much they changed to combine this paragraph with the earlier sections.
- L.369: please mention why these two levels are chosen. Are these inversion top and inversion base?
- L.371: it should be noted that the dropsondes were assimilated
- L.372: is this the absolute difference? Please clarify.
- L.378: mesoscale subsidence has been derived in George et al. (2023) and is part of the JOANNE dataset. It does not need to be speculated here. W4 seems to be from the last flight circle (Konow et al. (2021)). Looking at Fig. 1 of George et al. (2023) convergence exists in the afternoon flights at 2.5km. The claims should be in context to previous studies and here especially checked with the available data.
- L.380: "…revealed that the air above the cloud top descends along the downshear side of the cloud edge". Overlying in my head to position of W4 and the cloud top height product, W4 does not seem to be in the downstream location but rather right in the convective core. Indicating the dropsonde positions in e.g. Fig. 13 would help to make or dispute this claim more objectively.
- L.381: The radiative heating profiles of EUREC4A have been calculated by Albright et al. (2021) and should be checked if they support the radiative cooling hypothesis.
- L.426 ff: This paragraph is mostly literature review and proposes vague future analysis for aerosol studies which have only touched upon at the beginning of

this paragraph. To make the manuscript more concise this paragraph and all aerosol analysis can be removed.
- o L.465: without denial experiments this is speculation
- o L. 481: …The data measured by HALO are openly available at the …
- o L. 465: "conditions that favored the cloud development included the enhanced moisture around the flower and the increasing aerosol optical depth near Barbados":
  - Whether these conditioned favored the cloud development is speculation, these factors might just coexist by chance, especially so for the AOD
  - Could it not be that the Flower itself and their mesoscale circulations create these enhanced moisture structures?
- o L. 468: This outlook includes a lot of elements that are already used in this study, like aircraft measurements and in-situ measurements of cloud microphysics. What specifically are the authors missing and hope to find in a second revision of these observations. Couldn't those features be described in this manuscript so that a future study can focus solely on the dynamics? What are "interesting features"? This goes back to the question what the main goal and audience of this manuscript is.
- Figures
  - o Figure 1:
    - The synoptic chart with its fronts and coastlines does not match the underlying satellite image. Geographic features do not align.
    - Please indicate whether this is an AQUA or TERRA overpass
    - Please indicate the time of the overpass and the surface analysis chart
  - o Figure 2:
    - …GOES-16 ABI imagery… Please add ABI here and in all other figure captions where applicable
    - Mixture of m=milli and m=meter is confusing. For better comparison with models the radiances should be converted to brightness temperatures and included as a legend
    - The subplots change size. This should be pointed out or be adjusted.
  - o Figure 3:
    - …in Figure 2, and …
  - o Figure 4:
    - Please indicate whether this is UTC or local time
    - Unit and label are missing in colorbar
    - It should be made clear that the CTH is derived from satellite products and is therefore affected by attenuation. E.g. the frequencies after 14 are dropping at below 2km presumably due to attenuation of the layers above. This should be emphasized as the paper will be a resource for models.
  - o Figure 6:
    - Please improve the aspect ratio to 1:1 for longitude and latitude so that the cloud field is not skewed

- The colorbar ticks are not meaningful and too precise. It would help to interpret the satellite images better by marking typical cloud top heights at the lifting condensation level and inversion height.
- A sequential colormap would help aid the reader to better distinguish the heights and in particular their differences.
- Labels of the lat-lon grid are too small
- Turn off the interpolation of the observations to reduce the number of introduced artefacts.
- Caption: …cloud top height product between…
  - Figure 7:
    - It would be helpful to see the radiance threshold lines here as well, in particular to better follow the merging argument of the cloud cells
  - Figure 8:
    - Change colorbar positions so they match the image columns (e.g. switch optical depth and cloud top height)
    - Include radiance threshold outline
    - Switch off interpolation of the values here and in other figures where applicable
    - Please include the times of the snapshots on the left hand side to more easily find the timestamps of each satellite image.
  - Figure 9:
    - Caption: change "data" to "instrument" if this is true? Measured with data sounds incorrect.
  - Figure 10:
    - Color legend is missing.
  - Figure 11:
    - Is this a visible ABI image?
    - Increase font size of labels and ticks
    - This Figure could be combined with Fig. 13 by including the ATR track in each subfigure
    - Please include the position of the aircraft at the time when the satellite picture was taken.
  - Figure 12:
    - The caption is claiming that LWC is coming from the SAFIRE-CORE instrument, but this is in contradiction with the description in Bony et al. (2022)
    - SAFIRE-CORE is first mentioned here and not introduced beforehand. No reference is given.
    - Please add a horizontal line at a vertical velocity of 0 to better see where velocities are positive or negative.
    - Please write the time in the format HH:MM instead of MM:SS
  - Figure 13/14/15/16:
    - All these Figures can be combined to one and be presented similar to Fig. 8 or even a Fig. 8 continued.
    - The temporal frequency the satellite images are shown at could be reduced to 1h. I do not see the additional value of 30min snapshots.
  - Figure 18:

- Consider using km instead of m
- Please indicate the 700 and 750 hPa levels
- Figure 19:
  - Increase labels and ticks
- Figure 20:
  - Brackets in unit label should be removed
  - Consider using wind speed and wind direction as labels
- Figure 21:
  - …The dataset used is ERA5 at 0.25 deg resolution.
  - Units are missing
  - Consider using the term "Hovmöller diagram" instead of "longitudinal-temporal variations"
  - Please clarify whether the dataset is shown at a constant latitude or whether the central latitude of the cloud system is followed as well.

---

## Referee Comment (RC2)

**Review comments on**
**"Life cycle of a flower cloud system during the EUREC$^4$A field campaign" by Cui *et. al.**

**Summary and contributions**

The authors present a case study of the life cycle of an individual flower cloud system (FCS). The mesoscale FCS occurred in the tropical North Atlantic winter trades east of Barbados on 2 February 2020 during the EUREC$^4$A field campaign. The authors present new insights and outline favourable meteorological conditions suitable during the life cycle FCS. For example, the FCS was surrounded by an airmass having high relative humidity and showed increased aerosol optical depth. The FCS was identified by a radiance threshold of 102 mW m$^{-2}$ sr$^{-1}$ (cm$^{-1}$)$^{-1}$ in measurements of the GOES-16 ABI 11.2 $\mu$m channel and was manually tracked every 10 minutes. Based mainly on the total cloud area enclosed by the radiance threshold, the FCS life cycle was divided in (i) a developing, (ii) a mature, and (iii) a decaying stage. FCS characterization is based on cloud physical properties retrieved from GOES-16 ABI measurements, on cloud particle diameter measurements observed from the French SAFIRE ATR aircraft, and HALO aircraft dropsonde measurements. The environmental meteorological conditions have been studied with a variety of data, e.g. ERA5 reanalysis data, dropsonde measurements, retrievals from GOES-16 ABI as well as MODIS measurements. Studies about characterization and evolution of organized shallow convection and cloud organizations in the trades exist, to my knowledge however, the life cycle of an individual FCS was never considered. The results are sufficient to support the most interpretations/conclusions and they certainly will help evaluating numerical simulations. In my view, the presented work is worth publishing, as it provides meaningful contributions, most notably about the relationships between FCS life cycle stage and the meteorological (environmental) conditions. Even so, I believe some major improvements are needed before publication. Please find my detailed comments below.

**General Comments:**

1) The general impression is that the manuscript is written in a rush way and few ideas are mixed. English spelling is poor, but develops for the better as the manuscript progresses. A careful reading is necessary to smooth the whole document both the structure to there are many redundant sentences, the spelling, and miss-referencing of figures. May the authors ask a help from an native speaker at the department(s) for an assistance if necessary.

2) Data, methods, and results are mixed. Please structure the publication carefully. Temporal and spatial resolution is missing for any kind of data used and should be given in the data section. The description of the total precipitable water retrieval is missing in the data section. ERA5 is a powerful dataset that provides a great variety of meteorological variables, but which of them used for the study and their description is missing.

3) In my view, there are to many figures. The authors carefully should reduce the amount of figures. Move figures to appendix/supplementary material. Zoom on the FCS. Please show the $102$ mW m$^{-2}$ sr$^{-1}$ (cm$^{-1}$)$^{-1}$ isoline in any of the satellite images (radiance, CTH, COD) to clearly identify the FCS. Be consistent (UTC) vs. [UTC] or ($\mu$) vs. [microns]. Please use round brackets.

**Specific Comments**

1) line 93: The authors state they follow the tracking methodology of Fiolleau and Roca (2013). It is a automatic tracking algorithm and identification is based on image segmentation using five brightness temperature intervals. This is a contradiction to manual tracking based on one radiance threshold value. Please clarify.

2) lines 269-276: I wonder how the propagation speed was calculated. Which method was used to identify the arcs, etc pp. Please describe it in the methodology section.

3) line 351: The authors state that the increase in SST, which is only $\sim$0.3 K, plays an important role in increasing the boundary layer height, which allows for higher cloud top. Vial et al. (2019) considered LES as well as in-situ and satellite observations to conclude that SST and associated variations in sea-surface fluxes unlikely to play an important role in driving the diurnal cycle of marine trade-wind cumulus cloud cover in the Carribean. This is contradictory to your statement. Please clarify.

4) line 354: The authors make use of the sea-surface temperature and argue there is cold advection at the surface. This is misleading because the velocity of the sea surface, as is known, is different to the wind speed at 10 m above sea-surface. Would it not be better to use the 2 m temperature from ERA5?

**Technical Comments**

1) Please be consistent in wording. E.g. SAFIRE-CORE = French ATR aircraft = microphysiscs airborne platform (PMA) observations. And others ...

2) line 89: Can you please clarify what MCSs is.

3) line 119: Please give more information. To my knowledge CPSD is not a GOES-16 ABI retrieval.

4) Section 3 Synoptic pattern, line 146: ...a frontal system... Is the frontal system always associated with FCSs or is the pre-cold frontal area favourable for FCSs?

5) lines 154ff: This is methodology.

6) line 157: After a series of ... Can the authors describe why the chosen threshold is suitable for the identification of the FCS.

7) lines 165ff: This is methodology or maybe introduction.

8) line172: The life cycle is divided in three stages based mainly on the area of the FCS. The subsequent lines describe CTH is another constraint. Please indicate the constraints.

9) line 179: ITC $\longrightarrow$ UTC

10) line 179: ...at high levels $\longrightarrow$ do you mean at cirrus level? Please clarify.

11) line 188: I wonder what happens between 0500 UTC and 0900 UTC?

12) line 189: The authors state that the local maxima in CTH were related to convective activity. Isn't the develeopment of the FCS also related to convective activity? Please be specific.

13) line 190-191: ...can be seen in Figures 5h - 5i... I cannot see it. Do you mean Figures 6h-i?

14) line 193: expansion of growing cumulus $\longrightarrow$ Do you mean vertical growth?

15) line 195: What do you mean with 'linked by a linear cloud'? Do you mean 2 FCS form a cloud street, a cloud line or maybe shallow mesoscale overturning circulations (SMOCs; George, 2023).

16) line 196: The authors describe that a cloud region separated from the main cloud. Is the process of splitting and merging an important feature in the development of a FCS? Please clarify.

17) line 197: Please describe the term 'area of the detrainment/entrainment cloud region' in more detail

18) line 202/203: What do you mean with '... developed to higher levels and were not dominated by those with top heights lower than 1600 m...'?

19) line 204: ...some clouds... Do you mean convectively active parts of the FCS?

20) line 208: at 1220 UTC there is only one Figure $\longrightarrow$ 8.c5

21) line 213: brown color $\longrightarrow$ dark red

22) line 222: there is no Figure 9.b1 - b5

23) line 225/226: Please give a reference to guide the reader

24) lines 230-253: Figure references, e.g 8.d6 = 1240 UTC; Which type of distribution is shown in Figure 9 (absolute, relative, probability density)? What does #L/micron mean in Figure 9? What is the meaning of the dashed vertical lines in Figure 9? It could be worth to combine Figures 10 and 11. Please give at least a legend in Figure 10. What does mean volume diameter mean? Is it MVD = $2r_e$? Lines 239-242 refer to a height, but there is no height visible at all in Figure 10. Line 250/251 '...perhaps suggest...' $\longrightarrow$ shows

25) line 258: The authors state in section 4.1 the mature stage lasted until 1900 UTC.

26) line 259: The gust fronts ... Is it the area highligted by the pink arrow? Please clarify.

27) line 262/263: '...one region of precipitation seen in the satellite observation...' This is speculative because satellite imagery shows $r_e$ at cloud top. Furthermore, there is no Figure 7d9.

28) line 267: (Figure 13f) $\longrightarrow$ the pink arrow? Please clarify.

29) line 278: see comment line 258.

30) line 279/280: ...did not change much... I realize a wobbling patch.

31) line 281: ...a few local maxima of about 50 ... And some patches > 50.

32) line 285: ...the maximum values were about 45 $\mu$m ... decreasing to 45 $\mu$m...

33) line 293: Is there an explanation why $r_e$ is particularly high at the boundary of the FCS?

34) lines 296-302: Move to methodology section. Figure 17 can be combined with Figure 11.

35) lines 309-310: Please clarify which channel and quantity (radiance, brightness temp.) is shown. Focus in Fig. 19 on the FCS, e.g. decrease lon/lat.

36) lines 311/312: several times UTC is missing

37) line 345ff: there is no -(lon) °W. Check also the subsequent sections.

38) line 372: Differences are usually given in K.

39) line 386: Γd $\longrightarrow \Gamma_d$

40) line 403: The authors refer to cloud top height and pressure, but only the height is given. Please clarify.

41) line 410: ...during the developing and decaying stages... $\longrightarrow$ throughout the life cycle?

42) lines 412ff: move to data section.

43) line 430: MODIS is already defined.

44) line 480: EUREC4A $\longrightarrow$ EUREC$^4$A

45) line 481: (2021).. $\longrightarrow$ (2021)

46) line 485: ...propagation. analysis...

47) References: Please check alphabetical order.

48) page 22, Figure 1: Please indicate the time step of the surface weather analysis. To my knowledge the MODIS stripes are separated by about 100 minutes.

49) page 24, Figure 3: Shows the x-axis UTC or the time from evolution until decaying? Caption: describe the vertical lines.

50) page 25, Figure 4: Use the same tick marks on the x-axis as in Figure 3. Show the 3 stages in the life time of the FCS, e.g. vertical dotted lines. Show colorbar label (unit).

51) pages 26ff: Please revise the captions. Check (missing) colorbars $\longrightarrow$ labels/units. Figure 12 $\longrightarrow$ which time (UTC, local time); velocity unit missing; LWC unit missing. Figure 20a $\longrightarrow$ check unit.

52) Figures: be consistent with latitude and longitude. -59° = 59°W; latitude 10° = 10°N

---

## Author Comment (AC1)

**Response**

We thank the reviewer for reviewing our manuscript and providing very helpful and constructive suggestions. We have carefully considered all comments and have made changes to the manuscript. Please find below a detailed point-by-point response to all comments.

**General Comments:**
**1) The general impression is that the manuscript is written in a rush way and few ideas are mixed. English spelling is poor, but develops for the better as the manuscript progresses. A careful reading is necessary to smooth the whole document both the structure to there are many redundant sentences, the spelling, and miss-referencing of figures. May the authors ask a help from an native speaker at the department(s) for an assistance if necessary.**

All the above suggestions have been adopted. The revised manuscript is concise. All co-authors have read through the manuscript and are happy with this version.

**2) Data, methods, and results are mixed. Please structure the publication carefully. Temporal and spatial resolution is missing for any kind of data used and should be given in the data section. The description of the total precipitable water retrieval is missing in the data section. ERA5 is a powerful dataset that provides a great variety of meteorological variables, but which of them used for the study and their description is missing.**

We have restructured the manuscript by adding the methodology section and moving the relevant parts into the data section and method section as suggested.

The temporal and spatial resolutions have been added.

The total precipitable water has been deleted as another reviewer suggested.

The ERA5 description has been added.

**3) In my view, there are to many figures. The authors carefully should reduce the amount of figures. Move figures to appendix/supplementary material. Zoom on the FCS. Please show the 102 m W m$^{-2}$ (cm$^{-1}$)$^{-1}$ isoline in any of the satellite images (radiance, CTH, COD) to clearly identify the FCS.**

The number of figures has been reduced substantially. Figures 21-24 have been replaced by a single figure. Figures 25-26 have been deleted. Several other figures have been merged. The figure number has been reduced from 26 to 15.

The radiance has been converted to brightness temperature as suggested by another reviewer. The new Figure 7 shows the images with the threshold lines.

**Be consistent (UTC) vs. [UTC] or ( ) vs. [microns]. Please use round brackets.**

We have replaced the labels of Figures 9-10. They are now consistent.

**Specific Comments**

**1) line 93: The authors state they follow the tracking methodology of Fiolleau and Roca (2013). It is an automatic tracking algorithm and identification is based on image segmentation using ve brightness temperature intervals. This is a contradiction to manual**

**tracking based on one radiance threshold value. Please clarify.**

We did not using the tracking algorithm of Fiolleau and Roca (2013). Instead, we were inspired by their method. We have clarified in the revised version. "We follow their method to characterize the flower system on 2 February 2020." has been changed to "We were inspired by their method for detecting ing and tracking a cloud system using a brightness temperature threshold to characterize the flower system on 2 February 2020."

**2) lines 269-276: I wonder how the propagation speed was calculated. Which method was used to identify the arcs, etc pp. Please describe it in the methodology section.**

The following has been added in the methodology section.

A cloud arc associated with a cold pool moves due to a combination of factors, including its propagation, advection by the average winds within the cloud-bearing layers of its environment, and interactions with other cold pools. As a result, the true propagation speed of the cloud arc is the difference between its actual speed and the advection speed in cases where there are no interactions between cold pools. To determine the propagation speed, we first positioned the satellite image so that the flower was centred within the frame at each time. Next, we measured the distances between the cloud arcs at two subsequent times and used these measurements to estimate the propagation speed. Our speed estimation was limited to periods when the cloud arcs were well-defined and easily distinguishable.

**3) line 351: The authors state that the increase in SST, which is only 0.3 K, plays an important role in increasing the boundary layer height, which allows for higher cloud top. Vial et al. (2019) considered LES as well as in-situ and satellite observations to conclude that SST and associated variations in sea-surface fluxes unlikely to play an important role in driving the diurnal cycle of marine trade-wind cumulus cloud cover in the Carribean. This is contradictory to your statement. Please clarify.**

Sea surface temperature is a a controlling factor as discussed in the Introduction. We added the following in Section 6: Vial et al. (2019) considered Large Eddy Simulation (LES) data, as well as in-situ and satellite observations, and concluded that SST and its associated sea-surface flux variations are unlikely to be significant contributors to the diurnal cycle of marine trade-wind cumulus cloud cover in the Caribbean. However, Chen et al. (2023) explored the influence of minor SST anomalies at the submeso- to meso-scale level on the daily average of trade cumulus cloudiness, using satellite observations validated against ship-based measurements collected during the Atlantic Tradewind Ocean-Atmosphere Mesoscale Interaction Campaign (ATOMIC), conducted in conjunction with the EUREC[4]A field campaign. Their findings highlighted that localized variations in SST have the potential to influence daily cloud cover, with the primary driving force being the spatial diversity in surface-induced turbulence and surface heat flux, rather than the effects of surface or boundary layer convergence. Although the small change in SST of this case was unlikely of considerable importance in the development of the flower, the exact effect of SST is worth further investigation.

**4) line 354: The authors make use of the sea-surface temperature and argue there is cold advection at the surface. This is misleading because the velocity of the sea surface, as is known, is different to the wind speed at 10 m above sea-surface. Would it not be better to use the 2 m temperature from ERA5?**

We have deleted the figure containing the cold advection. As the reviewer suggested, number of figures has been reduced by combining and simplifying.

**Technical Comments**

**1) Please be consistent in wording. E.g. SAFIRE-CORE = French ATR aircraft = microphysiscs airborne platform (PMA) observations. And others …**

A consistent way of wording has been adopted as the reviewer suggested. The text has been changed accordingly.

**2) line 89: Can you please clarify what MCSs is.**

MCSs has been changed to "mesoscale convective systems (MCSs)".

**3) line 119: Please give more information. To my knowledge CPSD is not a GOES-16 ABI retrieval.**

The cloud particle size is an ABI Level 2 retrieval (Heidinger et al., 2020).

The following has been added in the description of COD and CPS.

The ABI Level 2 Cloud Particle Size (CPS) product provides the effective radius of the particles in a single cloud layer. Different algorithms are used for daytime and nighttime conditions. The retrieval of daytime cloud optical and microphysical properties is extensively explained in Walther and Heidinger (2012) and in the Algorithm Theoretical Basis Document by Walther and colleagues (2013). This methodology relies on measurements obtained from two distinct channels: one operating in the visible range as a non-absorbing window channel and another in the near-infrared part of the spectrum, which is weakly absorbing. Optical thickness is primarily determined through the visible channel's reflectance, but the absorption channel furnishes additional insights into absorber volume, enabling the estimation of an effective particle size. Furthermore, it indirectly aids in adjusting cloud optical depth (COD) estimates by accounting for variations in the phase function due to changes in forward-scattering direction resulting from differing particle sizes.

**4) Section 3 Synoptic pattern, line 146: ...a frontal system... Is the frontal system always associated with FCSs or is the pre-cold frontal area favourable for FCSs?**

"So-called flower clouds (Bony et al. 2020) were evident around 60 °W, near Barbados." has been changed to "So-called flower clouds (Bony et al. 2020; Schulz, 2022) were evident around 60 °W, near Barbados. It will be shown later that the flower cloud system studied in this paper developed from lines of convective clouds in the pre-cold frontal area."

**5) lines 154 : This is methodology.**

The paragraph in Lines 154-158 has been changed to "The detection of a cloud system is carried out by specifying a brightness temperature threshold on an IR image to mark the border of contiguous cloudy regions where the brightness temperature smaller than the threshold (e.g., Maddox, 1980). The threshold varies from case to case, in the range of 205 K to 233 K for mesoscale convective systems with deep convections (Maddox, 1980; Machado et al., 1998, Fiolleau and Roca, 2013). After a series of tests and trials, we found that a brightness temperature threshold of 290 K of GOES-16 IR channel 14 (11.2 μm) was a good choice to identify the flower system. The threshold is consistent with Figure S2 of Bony et al. (2020)."

This paragraph has been moved to the methodology section.

**6) line 157: After a series of ... Can the authors describe why the chosen threshold is suitable for the identification of the FCS.**

Please see the above, i.e., reply to comment on Line 154.

**7) lines 165 : This is methodology or maybe introduction.**

This paragraph has been moved to the methodology section.

**8) line172: The life cycle is divided in three stages based mainly on the area of the FCS. The subsequent lines describe CTH is another constraint. Please indicate the constraints.**

"based mainly on the area of the system" has been changed to "based mainly on the area and the CTH of the system.

**9) line 179: ITC -> UTC**

This has been corrected.

**10) line 179: ...at high levels -> do you mean at cirrus level? Please clarify.**

"The decaying stage was marked by a decreasing cloud area from 1900 ITC (Figure 3a) although some clouds still remained at high levels (Figure 4)." has been changed to "The decaying stage was marked by a decreasing cloud area from 19:00 ITC (Figure 3a) although some clouds still reached approximately 3 km above sea level (Figure 4)."

**11) line 188: I wonder what happens between 0500 UTC and 0900 UTC?**

More information can be found in the discussion of Figure 7.

**12) line 189: The authors state that the local maxima in CTH were related to convective activity. Isn't the development of the FCS also related to convective activity? Please be specific.**

"The local maxima in CTH (Figure 6h – 6j, and 6l) were related to convective activity." has been changed to "The local maxima in CTH (Figure 6h – 6j, and 6l) and the development of the system were related to convective activity."

**13) line 190-191: ...can be seen in Figures 5h - 5i... I cannot see it. Do you mean Figures 6h-i?**

"detrained cloudy air due to cumulus development can be seen in Figures 5h – 5i." Has been changed to "detrained cloudy air due to cumulus development can be seen in a region to the north-west of 56.5 °W and 12.5 °N in Figure 5i."

**14) line 193: expansion of growing cumulus -> Do you mean vertical growth?**

"...was caused by the expansion of growing cumulus" has been changed to "...was caused by the vertical development of cumulus in the area."

**15) line 195: What do you mean with 'linked by a linear cloud'? Do you mean 2 FCS form a cloud street, a cloud line or maybe shallow mesoscale overturning circulations (SMOCs; George, 2023).**

The new figure indicated that the outlines of the two cluster were not in contact. "Two main cloud clusters are shown in Figure 7a linked by a linear cloud as mentioned above." has been changed to "Two main cloud clusters had not yet merged as shown in Figure 7a."

**16) line 196: The authors describe a cloud region separated from the main cloud. Is the process of splitting and merging an important feature in the development of a FCS? Please clarify.**

"The splitting did not cause the drop in total area of the system during this period (Figure 3a), which implies that the splitting did not hinder the further development of the system." has been added after "A cloud region separated from the main cloud between 07:40 – 08:00 UTC as indicated by an arrow in Figures 7b-c".

**17) line 197: Please describe the term 'area of the detrainment/entrainment cloud region' in more detail**

"The area of the detrainment cloud region in the central cluster, as indicated by an arrow in Figure 7f, was seen to increase further in the next two hours." has been changed to "The area of the central cluster, as indicated by an arrow in Figure 7f, was seen to increase further in the next two hours."

**18) line 202/203: What do you mean with '... developed to higher levels and were not dominated by those with top heights lower than 1600 m...'?**

"and were not dominated by those with top height lower than 1600 m after the fast development (Figure 4)." has been deleted.

**19) line 204: ...some clouds... Do you mean convectively active parts of the FCS?**

"Some clouds developed to higher levels (Figures 8.b1-b5)." has been changed to "The convectively active parts of the system developed to higher levels (Figures 8.b1-b5)."

**20) line 208: at 1220 UTC there is only one Figure -> 8.c5**

This has been corrected.

**21) line 213: brown color -> dark red**

"(brown colour)" has been deleted since the figure was replotted using a different colour bar and merged with other figures as another reviewer suggested.

**22) line 222: there is no Figure 9.b1 – b5**

This has been corrected.

**23) line 225/226: Please give a reference to guide the reader**

However, there was significant local variability and indeed inversion tops determined from some dropsondes reached about 3100 m." has been changed to "However, there was significant local variability and indeed inversion tops determined from some dropsondes reached about 3100 m (Figure 14a in dropsonde analysis)."

**24) lines 230-253:**
**Figure references, e.g 8.d6 = 1240 UTC**

The figure has been replotted as another reviewer suggested. The time of each row has been added to the new figure.

**Which type of distribution is shown in Figure 9 (absolute, relative, probability density)?**

Figure 9 shows the drop size distribution function. See the reply to the next question.

**What does #L/micron mean in Figure 9?**

The drop number size distribution function is the number of particles per unit volume between size $D$ and $D + dD$. The unit is the drop number per volume (e.g., liter) per micrometer of size range. The $y$-axis label has been changed to "Concentration ($L^{-1}$ $\mu m^{-1}$)"

**What is the meaning of the dashed vertical lines in Figure 9?**

The dashed vertical lines have been removed.

**It could be worth to combine Figures 10 and 11.**

Figures 11 and 12 have been combined.

**What does mean volume diameter mean? Is it MVD = 2re?**

"... the median volume diameter (MVD), defined as the median of the cumulative mass size distribution, were derived from the composite size distributions" has been described in the data section.

**Lines 239-242 refer to a height, but there is no height visible at all in Figure 10.**

The following information has been added in Line 250.

The aircraft passed the first three clouds at 1100 m above sea level before ascending from 2040 m to 3060 m across the fourth cloud.

**Line 250/251 '...perhaps suggest...' → shows**

This has been corrected.

**25) line 258: The authors state in section 4.1 the mature stage lasted until 1900 UTC.**

This figure was used for both the mature and dissipating stages. Please note that another reviewer suggested to combine Figures 13-16 to form a single figure at a reduced frequency of one hour, which has been adopted. A new figure has been produced and replaced the figures. Figure numbers have been changed accordingly.

**26) line 259: The gust fronts ... Is it the area highligted by the pink arrow? Please clarify.**

The figure has been replotted as suggested by another reviewer and the interval has been reduced to 1 hour. It is now only one arrow in Figure 12a1.

**27) line 262/263: '...one region of precipitation seen in the satellite observation...' This is speculative because satellite imagery shows re at cloud top. Furthermore, there is no Figure 7d9.**

"There was only one region of precipitation seen in the satellite observations (Figure 7d9 for example)" has been changed to "There was only one region of effective radius greater than 40 μm (indicating the presence of precipitation) seen in the satellite observations (Figure 8d9) for example."

**28) line 267: (Figure 13f) -> the pink arrow? Please clarify.**

Please note that another reviewer suggested to combine Figures 13-16 to form a single figure at a reduced frequency of one hour, which has been adopted. The new figure uses red arrows. The text has changed to the new figure.

**29) line 278: see comment line 258.**

See response to Line 258.

**30) line 279/280: ...did not change much... I realize a wobbling patch.**

"did not change much" has been changed to "did not change significantly".

**31) line 281: ...a few local maxima of about 50 ... And some patches > 50.**

"than 30, with a few local maxima of about 50 appearing from time to time." has been changed to "than 30, with a few local maxima of about 50 and some patches > 50 occasionally"

**32) line 285: ...the maximum values were about 45 μm ... decreasing to 45 μm…**

", decreasing to 45 μm or less at 1600 UTC" has been deleted.

**33) line 293: Is there an explanation why re is particularly high at the boundary of the FCS?**

We have noticed the higher values of the effective radius at the edges of the flower. The possible reason is that overestimation in $r$e increases for scattering angles greater than 140˚ (Painemal et al., 2021). We did not cite Painemal et al. (2021) because they evaluated the GOES-13 imager rather than the GOES-16 ABI product. Our focus is the organised region of large $r_e$ in the central part of the flower which is unlikely affected by the large scattering angles as at the cloud edges.

**34) lines 296-302: Move to methodology section. Figure 17 can be combined with Figure 11.**

Lines 296-302 have been changed to "The HALO aircraft flew a clover pattern on 2 February (Konow et al. 2021). Those dropsondes allowed us to understand the temporal variation of the vertical stratification at certain locations. It took about 15 min for a dropsonde to fall from ~ 9 km into the boundary layer. We selected four dropsondes falling into or near the flower system. The temperature, relative humidity, and wind of the four dropsondes were examined in order to understand the temporal variations." It has been moved to the methodology section as the reviewer suggested.

Figure 17 has been deleted as another reviewer suggested. We have combined Figure 11 and Figure 12.

**35) lines 309-310: Please clarify which channel and quantity (radiance, brightness temp.) is shown. Focus in Fig. 19 on the FCS, e.g. decrease lon/lat.**

The caption has been changed to "Locations of the dropsondes imposed on radiance images of the GOES-16 ABI Band 6 for dropsonde W1-W4."

The domain has been reduced to focus on the flower system.

Please note that it is Figure 13 in the revised version.

**36) lines 311/312: several times UTC is missing**

UTC times have been added.

**37) line 345 : there is no -(lon) W. Check also the subsequent sections.**

This has been changed and all other occurrences have been checked and corrected.

**38) line 372: Differences are usually given in K.**

Figures 23, 25, and 26 have been removed.  Figures 21,22, and 24 have been simplified and merged to a single figure. The text in Section 6 has been rewritten. Therefore, Lines 355 to 444 were deleted. As a results, there are not responses to comments below on Lines 355 – 444.

**39) line 386: d -> d**

See reply to Line 372.

**40) line 403: The authors refer to cloud top height and pressure, but only the height is given. Please clarify.**

See reply to Line 372.

**41) line 410: ...during the developing and decaying stages... -> throughout the life cycle?**

See reply to Line 372.

**42) lines 412 : move to data section.**

See reply to Line 372.

**43) line 430: MODIS is already defined.**

See reply to Line 372.

**44) line 480: EUREC4A -> EUREC4A**

It has been corrected.

**45) line 481: (2021).. -> (2021)**

It has been corrected.

**46) line 485: ...propagation. Analysis…**

This has been changed to "propagation analysis."

**47) References: Please check alphabetical order.**

We have checked the order and made necessary changes.

**48) page 22, Figure 1: Please indicate the time step of the surface weather analysis. To my knowledge the MODIS stripes are separated by about 100 minutes.**

The caption of Figure 1 has been changed to "Figure 1: The MODIS Terra satellite image imposed with the surface pressure chart at 1200 UTC on 2 February 2020. The passover times of the satellite were approximately between 12:05 – 12:18, 13:45 – 13:57, and 15:23 – 15:36 UTC from east to west swaths, respectively."

**49) page 24, Figure 3: Shows the x-axis UTC or the time from evolution until decaying? Caption: describe the vertical lines.**

The x-axis label has been changed to Time (UTC). The vertical lines have been removed.

**50) page 25, Figure 4: Use the same tick marks on the x-axis as in Figure 3. Show the 3 stages in the life time of the FCS, e.g. vertical dotted line. Show colorbar label (unit).**

The new Figure 4 has been changed using the same tick marks on the x-axis as in Figure 3. The colour bar unit has been added.

**51) pages 26 : Please revise the captions. Check (missing) colorbars -> labels/units. Figure 12 -> which time (UTC, local time); velocity unit missing; LWC unit missing. Figure 20a -> check unit.**

The x-axis label has been changed to Time (UTC). The units of w and LWC have been added. The Unit in Figure 20 has been changed.

**52) Figures: be consistent with latitude and longitude. -59 = 59W; latitude 10 = 10N**

Yes, all the latitude and longitude have been checked and they are in a consistent way.

**References**

Heidinger, A. K., Pavolonis, M. J., Calvert, C., Hoffman, J., Nebuda, S., Straka, W., Walther, A., and Wanzong, S.: ABI Cloud Products from the GOES-R Series, in: The GOES-R Series: A New Generation of Geostationary Environmental Satellites, chap. 6, edited by: Goodman, S. J., Schmit, T. J., Daniels, J., and Redmon, R. J., Elsevier, 43–62, https://doi.org/10.1016/B978-0-12-814327-8.00006-8, 2020.

Painemal, D., Spangenberg, D., Smith Jr., W. L., Minnis, P., Cairns, B., Moore, R. H., Crosbie, E., Robinson, C., Thornhill, K. L., Winstead, E. L., and Ziemba, L.: Evaluation of satellite retrievals of

liquid clouds from the GOES-13 imager and MODIS over the midlatitude North Atlantic during the NAAMES campaign, Atmos. Meas. Tech., 14, 6633–6646, https://doi.org/10.5194/amt-14-6633-2021, 2021.

Walther, A. and Heidinger, A. K.: Implementation of the Daytime Cloud Optical and Microphysical Properties Algorithm (DCOMP) in PATMOS-x, J. Appl. Meteorol. Climatol., 51, 1371–1390, https://doi.org/10.1175/jamc-d-11-0108.1, 2012.

Walther, A., Straka, W., Heidinger, A.K., 2013. GOES-R Advanced Baseline Imager (ABI) Algorithm Theoretical Basis Document for Daytime Cloud Optical and Microphysical Properties (DCOMP), Version 3.0, 66 pp.

---

## Author Comment (AC2)

**Response**

We thank the reviewer for reviewing our manuscript and providing such thorough and constructive suggestions. We have considered the comments carefully and have made appropriate changes to the manuscript. Please find below a detailed point-by-point response to all comments.

**General Comments**

**- The manuscript would greatly improve by being more concise and more fluent. Paragraphs are often stand-alone and not well connected into the flow of the manuscript. An example is for example the 3rd and 4th paragraph in the introduction that mainly review cloud controlling factors and aerosols without being a strong part of the motivation of the manuscript.**

We revised the manuscript following the suggestion and comments. The controlling factor paragraph has been rewritten with inclusion of new studies from the EUREC$^4$A. The aerosol paragraph has been deleted as suggested.

**- Please gear towards modelers and emphasize that this is observational work. The manuscript should provide clear key parameters and their values. While this is a case study and cannot be generalized, key parameters of each stage in the life-cycle shall be summarized. This boils down to the question of what added-value does this manuscript provide if someone wants to set-up and compare their simulation with observations or study their dynamics more generally.**

It was always the intention to gear towards modellers and we thought this was clear. However, we appreciate your concern. It should be noted too that observations are limited, even with the many measurements made in EUREC$^4$A. We have edited the document and added information. The focus of the revised version is more specific. For example, the controlling factors are limited to the relevant ones. Besides, the variations of the cloud top height and the area of the flower can be used for the comparison between the modelling results and the observations. The development of the raindrops in the central part of the flower is a feature for modellers to compare.

**- The manuscript would greatly benefit by being carefully read by an English native speaker (e.g. one of the co-authors).**

All the co-authors have read through the manuscript and they are happy with the language and grammar.

**- Plenty of studies came out of the EUREC4A field campaign and several of these touch on aspects of this study as well, in particular Feb 2. The manuscript would greatly improve if it would embed previous results to strengthen its hypotheses. Examples are the mesoscale circulations and subsidence measured by George et al. (2023), the cold pool analysis by Touze-Pfeiffer et al. (2022) and the radiative cooling profiles by Albright et al. (2021). All of these studies present observations of Feb 2nd and are hardly mentioned in this manuscript. Cold pool dynamics that Dauhut et al. (2023) for example mentioned are described but hardly discussed in terms of their potential role although they seem to be essential for this pattern.**

The paragraphs between Lines 76 and 98 have been replaced by the following.

Klein et al. (2017) made a thorough review of low-cloud controlling factors. They found that a strengthened inversion stability reduces mixing across the inversion and keeps the boundary layer

shallower, more humid and cloudier. Mieslinger et al. (2019) also identified the atmospheric controls on the behaviour of shallow cumulus. They found that the surface wind speed was the most dominant control factor, much greater than others, such as lower-tropospheric stability, subsidence rate, sea surface temperature, total column water vapor, wind speed, wind shear, and Bowen ratio. The mesoscale organization of precipitating shallow cumulus was examined by Seifert and Heus (2013). They simulated the formation of cold pools due to evaporation of raindrops and found precipitation was required for aggregation. However, Bretherton and Blossey (2017) suggested that precipitation and mesoscale feedbacks are important but not essential for the self-aggregation of shallow cumulus. So the role of precipitation in aggregation is currently unclear.

The following has been added.

Several EUREC[4]A -based studies have focused on various aspects of shallow cumulus organization using information gathered during the field campaign, including the mesoscale organization, cold pool dynamics, radiative cooling profiles, and cloud cover variability. Narenpitak et al. (2021) studied the transition of trade cumulus organization on 2 February 2020 and emphasized the role of mesoscale total water convergence and large-scale upward motion. Dauhut et al. (2023) simulated the 2 February flower and found organized updraughts in an arc with the highest rain rates to the upwind side similar to a mesoscale convective system (MCS) with trailing stratiform. Touzé-Peiffer et al. (2022) developed a method to detect cold pools using atmospheric soundings and found that cold pools manifested temperature drops of $\sim 1$ K relative to the environment together with moisture increases of $\sim 1$ g kg$^{-1}$ and were accompanied by mesoscale cloud arcs. Albright et al. (2021) calculated radiative profiles from the HALO (High Altitude and Long Range Research Aircraft) soundings and highlighted the diurnal variations and individual variability associated with elevated moisture layers. George et al. (2023) found the existence of shallow mesoscale overturning circulations which influence subcloud layer drying efficiency, resulting in moist ascending and dry descending branches. Schulz and Stevens (2023) explored mesoscale variability in the trade-wind layer using large-eddy simulations and validated their results against EUREC[4]A observations. In their simulations, the simulated flowers did not appear as large round cloud systems but as scattered shallow cumuli like sugar clouds. They suggested that the primary cause lies in the inability of the simulations to reproduce the observed presence of stratiform layers associated with the flowers.

**- The aerosol section is very limited and less mature. I would encourage the authors to exclude any aerosol related analysis from this study.**

The aerosol section has been deleted.

**- Precipitation, which is closely connected to cold pools and has been pointed out to correlate with Flowers has been mostly ignored. An analysis of the precipitation, e.g., from satellite products or research vessels equipped with radar (e.g. Fig 8 of Touze-Pfeiffer et al. (2022)) should be included.**

We have carefully checked the radar dataset used in Touze-Pfeiffer et al. (2022). Their Figure 8 captured the only short period of precipitation on 2 February 2020. We also have explored the data availability of the ground-based radar, POLDIRAD. Unfortunately, the radar only operated after 2 February 2020. It is well known that satellites do not provide satisfactory precipitation retrievals of shallow cumulus clouds. The precipitation data used in the winter trades over the EUREC4A region is mainly from the Micro Rain Radar measurements (e.g., Schulz et al., 2021; Vial et al., 2021).

**Specific comments**

**- Consider capitalizing the mesoscale pattern names and make them italic for better readability**

The co-authors thought that it would be good to keep using names without capitalizing or italicizing them.

**- A reference to the PMA instrument suit and its measurements is missing in the data section.**

The paper, EUREC4A observations from the SAFIRE ATR42 aircraft, Earth Syst. Sci. Data, 14, 2021–2064, https://doi.org/10.5194/essd-14-2021-2022, 2022 by Bony et al., has been added. This paper has a section on PMA composites. The following has been added in the data section.

The French SAFIRE ATR aircraft flew mainly in the sub-cloud layer and near the cloud base level to carry out cloud microphysics, aerosol and many other atmospheric measurements. (Bony et al. 2022). Among the cloud instruments were the cloud droplet probe (CDP-2) and the 2D stereo probe (2D-S) on the Microphysics Airborne Platform (PMA dataset, Coutris 2021). The drop size distributions from the CDP-2 and 2D-s were combined to produce a single composite distribution from 2 μm to 2.55 mm with resolutions varying from $1-2$ μm up to 43 μm, through 10 μm up to 1 mm, to 100 μm from 1.05 to 2.55 mm. The liquid water content (LWC) and the median volume diameter (MVD), defined as the median of the cumulative mass size distribution, were derived from the composite size distributions.  Wind components were measured with a five-hole radome nose (SAFIRE-CORE dataset, TRAMM et al. 2021). More information on the observations, including the flight pattern and instrumentation, can be found in Bony et al. (2022).

**- Further, the manuscript mentions the SAFIRE-CORE dataset, which should be cited and explained in the data section as well.**

"The French SAFIRE ATR aircraft flew mainly in the sub-cloud layer and near the cloud base level to carry out cloud microphysics, aerosol and many other atmospheric measurements. The observations, including the flight pattern and instrumentation, from the ATR aircraft during the field campaign can be found in Bony et al. (2022)." has been changed to "The French SAFIRE ATR aircraft flew mainly in the sub-cloud layer and near the cloud base level to carry out cloud microphysics, aerosol and many other atmospheric measurements (Bony et al. 2022). Among the cloud instruments were the cloud droplet probe (CDP-2) and the 2D stereo probe (2D-S) on the Microphysics Airborne Platform (PMA dataset, Coutris 2021). The drop size distributions from the CDP-2 and 2D-s were combined to produce a single composite distribution from 2 μm to 2.55 mm with resolutions varying from $1-2$ μm up to 43 μm, through 10 μm up to 1 mm, to 100 μm from 1.05 to 2.55 mm. The liquid water content (LWC) and the median volume diameter (MVD), defined as the median of the cumulative mass size distribution, were derived from the composite size distributions.  Wind components were measured with a five-hole radome nose (SAFIRE-CORE dataset, TRAMM et al. 2021).. More information on the observations, including the flight pattern and instrumentation, can be found in Bony et al. (2022)."

**- Data availability section:**

**Please provide DOIs to the specific datasets used. URLs are not a permanent identifier and, in this case, do not point to the specific versions used.**
**DOIs of GOES-16 products used**
**DOI of SAFIRE-CORE dataset if it has been used as indicated in the manuscript**
**DOI of ERA5 hourly data product**
**Any additional datasets**

The DOIs have replaced the URLs except GOES 16 because GOES 16 datasets were moved to Amazon Web Service and the doi of this dataset could not been found. We have changed the data availability as follows: "GOES-16 data was accessed from https://registry.opendata.aws/noaa-goes. The MODIS bands 1 and 3 – 4 can be found at doi:10.5067/MODIS/MOD02QKM.061 and doi:10.5067/MODIS/MOD02HKM.061, respectively. The dropsonde dataset is available at https://doi.org/10.5281/zenodo.4746312. The SAFIRE-CORE and PMA/Cloud composite datasets from the SAFIRE ATR42 can be found at https://doi.org/10.25326/298 and https://doi.org/10.25326/237, respectively. The ERA5 datasets can be downloaded from the Copernicus Climate Change Service (C3S) Climate Data Store (CDS), DOI: 10.24381/cds.adbb2d47. All the above datasets were accessed on 07-11-2023."

**Code to reproduce results and figures is missing**

We have added the code availability in the revised version.

**- The number of figures needs to be drastically reduced. A few suggestions are proposed in the figure specific comments, but additional deletions/merges or moves to the supplement would benefit the focus of this manuscript.**

Figures 11-12 have been combined. Figures 18-20 have been merged. We have plotted a new figure which simplified and combined Figures 21-24. The four figures have been replaced by the new figure. Figures 25-26 have been deleted. In total, the figure number has been reduced from 26 to 15.

**- The manuscript is studying the life cycle of a cloud system within a Flowers pattern, which according to Vial et al. (2019) undergoes a diurnal cycle. However, the analysis here uses the foundation temperature of ERA5 which is lacking a diurnal cycle. Using the skin temperature might improve some of the robustness of arguments and should be used here instead. In-situ observations from ships would be even better.**

We have examined the skin temperature and it did not make much difference between the skin temperature and the SST patterns. The reason for using SST can be seen in the reply to "L. 351: What do change in boundary layer height due the authors expect from a change in sea surface temperature of 0.3K? It would be good to include here the CTH estimates from the satellite measurements and how much they changed to combine this paragraph with the earlier sections."

The ship observation only cover a short period of the flower life-cycle. Please see the reply to "Precipitation, which is closely connected to cold pools and has been pointed out to correlate with Flowers has been mostly ignored. An analysis of the precipitation, e.g., from satellite products or research vessels equipped with radar (e.g. Fig 8 of Touze-Pfeiffer et al. (2022)) should be included."

**- Dropsonde analysis section:**

**Here would be a good point to reference George et al. (2021)**

The following has been added in the data section, which cites the paper by George et al. (2021).

Dropsondes are useful to measure the vertical profiles of atmospheric parameters and to derive the areal-mean mass divergence from a series of dropsondes along a circular pattern over oceans during airborne field campaigns (e.g., Bony and Stevens, 2019). The same sampling strategy was adopted in the EUREC[4]A campaign (Konow et al., 2021). There were 76 dropsondes with good quality, and most of them were dropped along a circle of ~222 km in diameter and some of them fell within the circle. George et al. (2021) described the details of the Joint dropsonde Observations of the

Atmosphere in tropical North atlaNtic meso-scale Environments (JOANNE), including the type, the resolution and the performance of the dropsondes.

**L. 351: What do change in boundary layer height due the authors expect from a change in sea surface temperature of 0.3K? It would be good to include here the CTH estimates from the satellite measurements and how much they changed to combine this paragraph with the earlier sections. 17 dropsondes have been selected but only 4 were mentioned in the manuscript (W1 – W4). All information about grouping and soundings at N, E, S end of the circle can therefore be discarded, including Fig. 17.**

Only the dropsondes at the West have been included in the revised version. Figure 17 has been deleted.

**Fig. 18 c can be combined with Fig. 20, while Fig. 18 a,b,d can be deleted**

Figure 18c has been combined with Fig. 20, while Figures 18 a,b,d have been deleted.

**Line by line comments**

**L. 55: …inversion and keeps the boundary layer….**

"which reduces mixing across inversion keeps boundary layer shallower" has been deleted to focus only on the relevant factors.

**L. 58: please use the word cloudiness instead of cloud**

"increases cloud" has been deleted to focus only on the relevant factors.

**L. 64: Aerosol is an other controlling….**

The aerosol is no longer a part of this paper. Please see the reply to the general comments on aerosol. This paragraph has been removed.

**L. 70: The work of Mieslinger et al. (2019) would better fit into the previous paragraph and should be better integrated there.**

The paragraph on aerosol has been deleted. Please see the reply above.

The following has been added to the previous paragraph.

Mieslinger et al. (2019) also identified the atmospheric controls on the  behaviour of shallow cumulus. They found that the surface wind speed was the most dominant control factor, with a much greater influence than other factors, such as lower-tropospheric stability, subsidence rate, sea surface temperature, total column water vapor, wind speed, wind shear, and Bowen ratio.

**L.86: …edge only of one Flower cloud patch with the highest…**

"edge only of the flower cloud with the highest rain rates and the largest cloud liquid water content" has been deleted to focus only on the relevant factors.

**L. 95: The Trade wind Alley is a jargon word that seems to be better described here by "downstream trades"**

"The Trade wind Alley" is the terminology used in overview paper of EUREC[4]A by Stevens et al. (2021). However, this has been replaced by downstream trades as the reviewer suggested.

**L. 101: "the life cycle of an individual flower system has not been examined". While this is true, the work of Dauhut et al. (2023) and Narenpitak et al. (2021) should definitely be mentioned here. It should also be emphasized that this study uses observations.**

A paragraph has been added above the line on the EUREC[4]A -based studies.

**L.102: The outlook to "investigate favorable and unfavorable conditions for this flower system" is not given in the manuscript. Only one cloud system is observed and neither sensitivity runs are made nor are a significant number of cloud systems observed to correlate the cloud pattern with cloud controlling factors.**

"investigate favorable and unfavorable conditions for this flower system" has been changed to "investigate environmental conditions along the trajectory of the system using reanalysis data"

**L.108: Switch "Geostationary Operational Environmental Satellite-16 (GOES-16)" with "Advanced Baseline Imager (ABI)". It's the imager which provides images and not the satellite itself.**

It has been switched.

**l. 115: please add a reason for why band 14 has been used which is not in the clean infrared window but is affected by water vapor.**

The air above the marine boundary layer in this case was very dry. The relative humidity was below 10% at 700 hP and about 20% at 750 hPa. The moisture impact was insignificant. The brightness temperature threshold of 290 K using this band for plotting Figure 2 produced the same result as in Figure S2 of Bony et al. (2020) using the other band.

**l.118: please provide a reference for the mentioned algorithms**

The following has been added in the data section.

Cloud-top properties are retrieved from the ABI sensor onboard GOES-16 using the GOES-R Algorithm Working Group (AWG) cloud height algorithm (ACHA) as described by Heidinger et al. (2020). The baseline ACHA simultaneously estimates cloud-top temperature, cloud emissivity at 11.2 μm, and a cloud microphysical index. Cloud-top height (CTH) is derived from cloud-top temperature using numerical weather prediction profiles. Although the baseline ACHA product has been thoroughly validated (Vaughan et al., 2009), it is still subject to errors associated with the assumption of constant emissivity, the assumption of a thin cloud layer, the presence of a lower cloud layer,  an inaccurate estimate of the surface temperature, an inaccurate estimate of the temperature profile, and/or instrument noise (Menzel et al., 2015). The uncertainty is noticeable when clouds are optically thin, and/or the underlying opaque layer is in the middle troposphere, and/or the surface temperature estimate does not track surface temperature change.

The ABI Level 2 Cloud Particle Size (CPS) product provides the effective radius of the particles in a single cloud layer. Different algorithms are used for daytime and nighttime conditions. The retrieval of daytime cloud optical and microphysical properties is extensively explained in Walther

and Heidinger (2012) and in the Algorithm Theoretical Basis Document by Walther et al. (2013). This methodology relies on measurements obtained from two distinct channels: one operating in the visible range as a non-absorbing window channel and another in the near-infrared part of the spectrum, which is weakly absorbing. Optical thickness is primarily determined through the visible channel's reflectance, but the absorption channel furnishes additional insights into absorber volume, enabling the estimation of an effective particle size. Furthermore, it indirectly aids in adjusting cloud optical depth (COD) estimates by accounting for variations in the phase function due to changes in forward-scattering direction resulting from differing particle sizes.

**l.137: The circling strategy of the EUREC4A flights has not been used here and is unimportant for this manuscript. Please leave this out and rather explain how the inversion base and inversion top are derived from the dropsonde dataset.**

Lines 136-139 have been deleted. For the explanation on how the inversion base and inversion top are derived from the dropsonde dataset, please see the reply to "L. 305: how are inversion top and inversion base derived from the dropsonde data? Please add this information to the data section."

**L.137: George et al. (2021) seems to be a better reference to the JOANNE dropsonde dataset. If JOANNE has not been used here, it should be justified.**

George et al. (2021) has been referenced. Please see the reply to Dropsonde analysis section: Here would be a good point to reference George et al. (2021).

**L.142: …aerosol and among other atmospheric measurements….**

This paragraph has been changed to "The French SAFIRE ATR aircraft flew mainly in the sub-cloud layer and near the cloud base level to carry out cloud microphysics, aerosol and many other atmospheric measurements. Among the cloud instruments were the cloud droplet probe (CDP-2) and the 2D stereo probe (2D-S) on the Microphysics Airborne Platform (PMA). The drop size distributions from the CDP-2 and 2D-s were combined to produce a single composite distribution from 2 µm to 2.55 mm with resolutions varying from $1 - 2$ µm up to 43 µm, through 10 µm up to 1 mm, to 100 µm from 1.05 to 2.55 mm. The liquid water content (LWC) and the median volume diameter (MVD), defined as the median of the cumulative mass size distribution, were derived from the composite size distributions. Wind components were measured with a five-hole radome nose. More information on the observations, including the flight pattern and instrumentation, can be found in Bony et al. (2022)."

**L.151: The C3ONTEXT dataset of Schulz (2022) could serve as a reference here to support the presence of a Flowers pattern.**

This paper has been cited. The sentence has been changed to "So-called flower clouds (Bony et al. 2020; Schulz, 2022) were evident around 60 °W, near Barbados."

**L.158: …GOES-16 ABI IR channel… it is the IR channel of the ABI and not GOES-16. Please adapt all occurrences in the script.**

The Channels are described in the Datasets.

For clarification, ABI has been added in all occurrences in the revised manuscript.

**L. 159: see above**

Added.

**L.158: brightness temperature would be more informative and intuitive then radiance, especially with respect to model simulations. Please convert the radiances in the script to brightness temperatures.**

Yes, the radiance has been converted to brightness temperature for the IR images. Figure 2 has been replotted using the brightness temperature.

**L.168: "Other studies used modified…". Unimportant references as only the Houze (2004) will be used. Consider writing "We adapted this system here under the assumption that Flowers are MCSs following Dauhut et al. (2023)"**

"Other studies used modified stages, for example, four stages ( convective initiation, genesis, mature, and decay) by Coniglio et al. (2010), and five stages (two growing stages (40% and 80% of its maximum area), a mature stage, and two dissipating stage (80% and 40% of its maximum area)) by Wall et al. (2018)." has been deleted.

**L.171: …divided into those three stages and based mainly on the area of the system."**

Corrected.

**L. 178: … and highest inversion top (~3100m) was determined by…**

Corrected.

**L.179: 19 UTC**

Corrected.

**L.196: 0720 – 0820 UTC (separation is not visible at 0800 UTC)**

Corrected.

**L.197: Figures 7a-7d).**

Corrected.

**L.198: …The area to the east…**

It is indeed the area of the east cluster.

**L. 199: "the two cloud regions merged at about 11 UTC." This is vague and could be more specific. The clouds seem clustered already in 7a s there is no clear-sky separating them. Maybe one could say that by 11UTC a common stratocumulus has been developed.**

"the two cloud regions merged at about 11 UTC" has been changed to "By around 11 UTC the stratiform in the detrainment layer of the two clouds had developed."

**L. 215: Units are weirdly formatted here and throughout the manuscript. Please use the same font for µ as for other characters.**

Yes, all characters are formatted in the same way in the revised version.

**L. 240: the "mean volume diameter" needs to be better defined as well as the entire PMA observations. A good place for this would be the data section.**

The following has been added in the data section.

The French SAFIRE ATR aircraft flew mainly in the sub-cloud layer and near the cloud base level to carry out cloud microphysics, aerosol and many other atmospheric measurements. (Bony et al. 2022). Among the cloud instruments were the cloud droplet probe (CDP-2) and the 2D stereo probe (2D-S) on the Microphysics Airborne Platform (PMA dataset, Coutris 2021). The drop size distributions from the CDP-2 and 2D-s were combined to produce a single composite distribution from 2 μm to 2.55 mm with resolutions varying from $1 - 2$ μm up to 43 μm, through 10 μm up to 1 mm, to 100 μm from 1.05 to 2.55 mm. The liquid water content (LWC) and the median volume diameter (MVD), defined as the median of the cumulative mass size distribution, were derived from the composite size distributions. Wind components were measured with a five-hole radome nose (SAFIRE-CORE dataset, TRAMM et al. 2021). More information on the observations, including the flight pattern and instrumentation, can be found in Bony et al. (2022).

**L. 250: where does the vertical velocity and LWC come from? Please define this in the data section.**

See the reply above (i.e., L240).

"The core data of SAFIRE ATR42 data can be found at https://doi.org/10.25326/298" has been added in the data availability.

**L. 250: at which altitude did the airplane sample these measurements? Was the flight level constant (above sea level or pressure)?**

The following information has been added.

The aircraft passed the first three clouds at 1100 m above sea level before ascending from 2040 m to 3060 m across the fourth cloud.

**L. 250: Figure 12 has been taken at 15 UTC, when the ATR has finished n2, which let me conclude that the highest values of LWC and upward motion were recorded at the WESTERN side of the cloud system, which would be in agreement with Dauhut et al. (2023).**

Figure 12 shows the MVD in n4 was smaller than that in n3, but the LWC values of n3 and n4 were similar (Figure 12), which indicates that the precipitation in n4 was much weaker although the sampling in n4 was taken at higher altitude. The updraughts in n4 were much weaker compared with in $n1 - n3$. It is hard to say if the observation is in agreement with Dauhut et al. (2023).

**L. 251: this is too vague.**

It has been changed to "The observations show a different picture than the one of a classical mesoscale convective system described by the LES (Large Eddy Simulation) modelling study of Dauhut et al (2023) where the small-scale cumulus clouds were found on the western side of the cloud system only and not throughout much of the area of the system."

**L. 305: how are inversion top and inversion base derived from the dropsonde data? Please add this information to the data section.**

The following paragraph has been added in the methodology section.

The vertical structure of the trade-wind regions is mainly characterized by a sharp inversion in temperature at the boundary layer top. The inversion base was manually examined and determined as the bottom of the major layer in which the temperature increases with altitude, whilst the inversion top as the bottom of the layer just above the inversion in which the temperature decreased with altitude.

**L. 351: What do change in boundary layer height due the authors expect from a change in sea surface temperature of 0.3K? It would be good to include here the CTH estimates from the satellite measurements and how much they changed to combine this paragraph with the earlier sections.**

As stated by Desbiolles et al. (2018), mesoscale SST variability plays an important role in shaping local atmospheric boundary layers through thermodynamic processes. However, it is very difficult to estimate the changes in the marine boundary layer height (MBLH) due to SST changes because the MBLH is also affected by other factors or processes, for example, the co-variation of surface wind speed and sea surface temperature (He and Lin, 2019), and the decoupling of the boundary layer (Wood and Bretherton,2004). An estimation requires a model with good representation of all processes across the surface and within and above the boundary layer (e.g., Sullivan et al., 2021), which is beyond the scope of this paper. However, Chen et al. (2023) explored the influence of minor SST anomalies at the submeso- to meso-scale level on the daily average of trade cumulus cloudiness. Their investigation relied on satellite observations, which were validated against ship-based measurements collected during the Atlantic Tradewind Ocean-Atmosphere Mesoscale Interaction Campaign (ATOMIC), conducted in conjunction with the EUREC⁴A field campaign. Their findings highlighted that localized variations in SST have the potential to influence daily cloud cover, with the primary driving force being the spatial diversity in surface-induced turbulence and surface heat flux, rather than the effects of surface or boundary layer convergence.

Figures 23, 25, and 26 have been removed. Figures 21,22, and 24 have been simplified and merged to a single figure. The text in Section 6 has been rewritten. The sentences starting from Chen et al. (2023) has been used in the new Section 6.

**L.369: please mention why these two levels are chosen. Are these inversion top and inversion base?**

Figures 23, 25, and 26 have been removed. Figures 21,22, and 24 have been simplified and merged to a single figure. The text in Section 6 has been rewritten. Therefore, Lines 355 to 444 were deleted. As a result, there are not responses to comments below on Lines 355 – 444.

**L.371: it should be noted that the dropsondes were assimilated**

Please see reply to Line 369.

**L.372: is this the absolute difference? Please clarify.**

Please see reply to Line 369.

**L.380: "…revealed that the air above the cloud top descends along the downshear side of the cloud edge". Overlying in my head to position of W4 and the cloud top height product, W4 does not seem to be in the downstream location but rather right in the convective core.**

**Indicating the dropsonde positions in e.g. Fig. 13 would help to make or dispute this claim more objectively.**

Please see reply to Line 369.

**L.381: The radiative heating profiles of EUREC4A have been calculated by Albright et al. (2021) and should be checked if they support the radiative cooling hypothesis.**

Please see reply to Line 369.

The net radiative cooling of W4 has been plotted. It supports the radiative cooling hypothesis.

**L.426 ff: This paragraph is mostly literature review and proposes vague future analysis for aerosol studies which have only touched upon at the beginning of this paragraph. To make the manuscript more concise this paragraph and all aerosol analysis can be removed.**

Please see reply to Line 369.

**L.465: without denial experiments this is speculation**

Because of the reduction in figures and shortened as suggested by the reviewer, the discussion on AOD and TPW has been removed. The sentence "Other conditions that favoured the cloud development included the enhanced moisture around the flower and the increasing aerosol optical depth near Barbados." has been deleted.

**L. 481: …The data measured by HALO are openly available at the …**

In the data availability section, the HALO part has been changed to "The dropsonde dataset is available at https://doi.org/10.5281/zenodo.4746312." Please see the reply to the Data Availability above.

**L. 465: "conditions that favored the cloud development included the enhanced moisture around the flower and the increasing aerosol optical depth near Barbados":**

**Whether these conditioned favored the cloud development is speculation, these factors might just coexist by chance, especially so for the AOD**

**Could it not be that the Flower itself and their mesoscale circulations create these enhanced moisture structures?**

The sentence has been deleted. Please see the reply to "Line 465: without denial experiments this is speculation".

**L. 468: This outlook includes a lot of elements that are already used in this study, like aircraft measurements and in-situ measurements of cloud microphysics. What specifically are the authors missing and hope to find in a second revision of these observations. Couldn't those features be described in this manuscript so that a future study can focus solely on the dynamics?**

The paragraph has been changed to "Future analysis will help to understand the internal structure of the flower and the microphysical processes in the life cycle of the flower. Aircraft measurements and radar observations may help to clarify some features which cannot be generalized from a case

study alone reveal some interesting features. For example, whether did the flower system operated as a large cloud in the mature stage and whether the cloud arcs were produced by one strong downdraught in the dissipating stage?"

**What are "interesting features"? This goes back to the question what the main goal and audience of this manuscript is.**

It has been changed to "Aircraft measurements and radar observations may help to clarify some features which cannot be generalized from a case study alone. For example, did the flower system operated as a large cloud in the mature stage and were the cloud arcs were produced by one strong downdraught in the dissipating stage?"

**Figures**

**Figure 1:**
**The synoptic chart with its fronts and coastlines does not match the underlying satellite image. Geographic features do not align. Please indicate whether this is an AQUA or TERRA overpass Please indicate the time of the overpass and the surface analysis chart**

The figure has been rearranged so that the coastlines match the underlying satellite image.

It was the TERRA overpass and we have added it and its overpass times as well as the surface chart time in the caption.

**Figure 2:**
**…GOES-16 ABI imagery… Please add ABI here and in all other figure captions where applicable Mixture of millimetre and micronmeter is confusing. For better comparison with models the radiances should be converted to brightness temperatures and included as a legend**

The radiance has been converted to brightness temperature, and a new figure of the brightness temperature has replaced the old one.

**The subplots change size. This should be pointed out or be adjusted.**

Yes, we have changed the caption of the figure.

Figure 2: Time sequence of GOES-16 ABI imagery of the infrared 11.2 μm band every three hour from 0300 to 1800 UTC on 2 February 2020. The red curve represents the brightness temperature of 290 K. Please note the change in subplot size.

**Figure 3:**
**…in Figure 2, and …**

We have changed the caption to "Figure 3: Temporal variation of (a) the area of the cloud system which is defined by the enclosed area with the red curve in Figure 2, and (b) the mean cloud top height of the cloud system."

**Figure 4:**
**Please indicate whether this is UTC or local time**

The x-axis label has been changed to UTC.

**Unit and label are missing in colorbar**

The unit and label have been added in the new figure.

**It should be made clear that the CTH is derived from satellite products and is therefore affected by attenuation. E.g. the frequencies after 14 are dropping at below 2km presumably due to attenuation of the layers above. This should be emphasized as the paper will be a resource for models.**

The sentence "Cloud top height (CTH) is retrieved with an algorithm using ABI infrared bands." has been replaced by "Cloud-top properties are retrieved from the ABI sensor onboard GOES 16 using the GOES-R Algorithm Working Group (AWG) cloud height algorithm (ACHA). The baseline ACHA simultaneously estimates cloud-top temperature, cloud emissivity at 11.2 μm, and a cloud microphysical index. Cloud-top height is derived from cloud-top temperature using numerical weather prediction profiles. Although the baseline ACHA product has been thoroughly validated (Vaughan et al., 2009), they are still subject to errors associated with the assumption of constant emissivity, the assumption of a thin cloud layer, the presence of a lower cloud layer, an inaccurate estimate of the surface temperature, an inaccurate estimate of the temperature profile, and/or instrument noise (Menzel et al., 2015). The uncertainty is noticeable when clouds are optically thin, and/or the underlying opaque layer is in the middle troposphere, and/or the surface temperature guess does not track surface temperature change."

**Figure 6:**
**Please improve the aspect ratio to 1:1 for longitude and latitude so that the cloud field is not skewed The colorbar ticks are not meaningful and too precise. It would help to interpret the satellite images better by marking typical cloud top heights at the lifting condensation level and inversion height. A sequential colormap would help aid the reader to better distinguish the heights and in particular their differences. Labels of the lat-lon grid are too small Turn off the interpolation of the observations to reduce the number of introduced artefacts.**

The figure has been replotted so that aspect ratio is 1:1. The colourbar has been changed to 500 – 400 m with an interval of 500 m. Both the lifting condensation level and the inversion height are functions of time, longitude, and latitude. Unfortunately, we do not have the data to calculate those levels. The labels have been increased.

**Caption: …cloud top height product between…**
Figure 6: Time sequence of GOES-16 ABI cloud top height product between 0000-1100 UTC. The region is between 60 °W – 54 °W and 12 -14 °N.

**Figure 7:**
**It would be helpful to see the radiance threshold lines here as well, in particular to better follow the merging argument of the cloud cells**

The figure has been replotted, and the brightness temperature threshold lines have been added. The aggregation can be seen in this figure.

**Figure 8:**
**Change colorbar positions so they match the image columns (e.g. switch optical depth and cloud top height) Include radiance threshold outline Switch off interpolation of the values here and in other figures where applicable. Please include the times of the snapshots on the left hand side to more easily find the timestamps of each satellite image.**

The figure has been replotted. The colourbars were added in each panel. The times of the snapshots have been added on the right hand side because the y-axis labels were on the left hand side.

The threshold outline was added in Figure 7. Adding the threshold lines caused the figure to become too messy, since the size of the 36 subplots were not big enough. Therefore, the threshold outlines were not included in this figure.

**Figure 9:**
**Caption: change "data" to "instrument" if this is true? Measured with data sounds incorrect.**

It has been changed to "Figure 9: The averaged drop size distribution measured with the PMA microphysical instrument from the ATR aircraft."

**Figure 10:**
**Color legend is missing.**

It has been added.

**Figure 11:**
**Is this a visible ABI image?**
**Increase font size of labels and ticks**
**This Figure could be combined with Fig. 13 by including the ATR track in each subfigure**
**Please include the position of the aircraft at the time when the satellite picture was taken.**

It is an ABL image and this has been indicated in the new caption.

The label size and the ticks have been increased.

Figures 11 and 12 have been merged.

Figure 11: (a) GOES-16 ABI image at 1500 UTC with the ATR flight tracks when the four clouds, represented as coloured curve in panel (b), were passed; and (b) the vertical velocity (unit: m s$^{-1}$) and the LWC (unit: g m$^{-3}$) at 1 Hz from SAFIRE--ATR42 between 14:53:08 and 15:18:58 UTC.

**Figure 12:**
**The caption is claiming that LWC is coming from the SAFIRE-CORE instrument, but this is in contradiction with the description in Bony et al. (2022) SAFIRE-CORE is first mentioned here and not introduced beforehand. No reference is given.**

It has been changed to SAFIRE—ATR42.

**Please add a horizontal line at a vertical velocity of 0 to better see where velocities are positive or negative.**

A horizontal line of 0 has been added to the vertical velocity plot.

**Please write the time in the format HH:MM instead of MM:SS**

The time has been written in HH:MM.

**Figure 13/14/15/16:**

**All these Figures can be combined to one and be presented similar to Fig. 8 or even a Fig. 8 continued.**

Figures 13 – 16 have been combined as a new figure similar to Figure 8.

**The temporal frequency the satellite images are shown at could be reduced to 1h. I do not see the additional value of 30min snapshots.**

Figures 13-16 have been combined and the interval has been reduced to one hour.

**Figure 18:**
**Consider using km instead of m. Please indicate the 700 and 750 hPa levels**

Figures 18 and 20 have been combined. The unit of y-axis has been changed to km and the horizontal lines have added to indicate the 750 and 700 hPa Levels.

**Figure 19:**
**Increase labels and ticks**

Labels and ticks have been increased.

**Figure 20:**
**Brackets in unit label should be removed**

Figures 18 and 20 have been combined. The brackets have been deleted.

**Consider using wind speed and wind direction as labels**

Wind speed and wind direction have been used in the new figure.

**Figure 21:**
**…The dataset used is ERA5 at 0.25 deg resolution.**
**Units are missing**
**Consider using the term "Hovmöller diagram" instead of "longitudinal-temporal variations"**
**Please clarify whether the dataset is shown at a constant latitude or whether the central latitude of the cloud system is followed as well.**

As the reviewer suggested, we have condensed Figures 21-24 in Section 6 to a single figure which shows the atmospheric conditions along the trajectory of the flower. There is no Hovmöller diagram in the revised version.

**References**

Bony, S., Lothon, M., Delanoë, J., Coutris, P., Etienne, J.-C., Aemisegger, F., Albright, A. L., André, T., Bellec, H., Baron, A., Bourdinot, J.-F., Brilouet, P.-E., Bourdon, A., Canonici, J.-C., Caudoux, C., Chazette, P., Cluzeau, M., Cornet, C., Desbios, J.-P., Duchanoy, D., Flamant, C., Fildier, B., Gourbeyre, C., Guiraud, L., Jiang, T., Lainard, C., Le Gac, C., Lendroit, C., Lernould, J., Perrin, T., Pouvesle, F., Richard, P., Rochetin, N., Salaün, K., Schwarzenboeck, A., Seurat, G., Stevens, B., Totems, J., Touzé-Peiffer, L., Vergez, G., Vial, J., Villiger, L., and Vogel, R.: EUREC4A observations from the SAFIRE ATR42 aircraft, Earth Syst. Sci. Data, 14, 2021–2064, https://doi.org/10.5194/essd-14-2021-2022, 2022.

Chen, X., Dias, J., Wolding, B., Pincus, R., DeMott, C., Wick, G., Thompson, E. J., and Fairall, C. W.: Ubiquitous Sea Surface Temperature Anomalies Increase Spatial Heterogeneity of Trade-Wind Cloudiness on Daily Timescale. *J. Atmos. Sci.*, https://doi.org/10.1175/JAS-D-23-0075.1, in press, 2023.

Desbiolles, F, Blamey, R, Illig, S, *et al.* Upscaling impact of wind/sea surface temperature mesoscale interactions on southern Africa austral summer climate. *Int J Climatol*. 38, 4651–4660. https://doi.org/10.1002/joc.5726, 2018.

Hersbach, H., Bell, B., Berrisford, P., Biavati, G., Horányi, A., Muñoz Sabater, J., Nicolas, J., Peubey, C., Radu, R., Rozum, I., Schepers, D., Simmons, A., Soci, C., Dee, D., Thépaut, J-N. (2023): ERA5 hourly data on single levels from 1940 to present. Copernicus Climate Change Service (C3S) Climate Data Store (CDS), DOI: 10.24381/cds.adbb2d47 (Accessed on DD-MMM-YYYY).

Menzel, W., Frey, R., and Baum, B.: Cloud Top Properties and Cloud Phase – Algorithm Theoretical Basis Document, Tech. rep., University of Wisconsin – Madison, 73 pp., available at:https://modis-images.gsfc.nasa.gov/_docs/MOD06-ATBD_2015_05_01.pdf (last access: 31 October 2023), 2015.

Mieslinger, T., Horváth, Á., Buehler, S. A., and Sakradzija, M.: The Dependence of Shallow Cumulus Macrophysical Properties on Large-Scale Meteorology as Observed in ASTER Imagery, J. Geophys. Res.-Atmos., 124, 11477–11505, 2019.

Schulz, H., Eastman, R., and Stevens, B.: Characterization and evolution of organized shallow convection in the trades, J. Geophys. Res.-Atmos., 126, e2021JD034575, https://doi.org/10.1029/2021JD034575, 2021.

Schulz, H.: C3ONTEXT: a Common Consensus on Convective OrgaNizaTion during the EUREC4A eXperimenT, Earth Syst. Sci. Data, 14, 1233–1256, https://doi.org/10.5194/essd-14-1233-2022, 2022.

Sullivan, P. P., J. C. McWilliams, J. C. Weil, E. G. Patton, and H. J. S. Fernando: Marine Boundary Layers above Heterogeneous SST: Alongfront Winds. *J. Atmos. Sci.*, **78**, 3297–3315, https://doi.org/10.1175/JAS-D-21-0072.1, 2021.

Vaughan, M. A., Powell, K. A., Kuehn, R. E., Young, S. A., Winker, D. M., Hostetler, C. A., Hunt, W. H., Liu, Z., McGill, M. J., and Getzewich, B. Z.: Fully automated detection of cloud and aerosol layers in the CALIPSO lidar measurementsMieslinger, J. Atmos. Ocean. Tech., 26, 2034–2050, https://doi.org/10.1175/2009JTECHA1228.1, 2009.

Vial, J., Vogel, R., and Schulz, H.: On the daily cycle of mesoscale cloud organization in the winter trades, Q. J. Roy. Meteorol. Soc., 47, 2850–2873, https://doi.org/10.1002/qj.4103, 2021.

Wood, R. and Bretherton, C. S.: Boundary layer depth, entrainment and decoupling in the cloud-capped subtropical and tropical marine boundary layer, J. Clim., 17, 3576–3588, 2004.